# Alkylative damage of mRNA leads to ribosome stalling and rescue by *trans* translation in bacteria

**Erica N Thomas, Kyusik Q Kim, Emily P McHugh, Thomas Marcinkiewicz, Hani S Zaher\***

Department of Biology, Washington University in St. Louis, St. Louis, United States

**Abstract** Similar to DNA replication, translation of the genetic code by the ribosome is hypothesized to be exceptionally sensitive to small chemical changes to its template mRNA. Here we show that the addition of common alkylating agents to growing cultures of *Escherichia coli* leads to the accumulation of several adducts within RNA, including N(1)-methyladenosine ($m^1A$). As expected, the introduction of $m^1A$ to model mRNAs was found to reduce the rate of peptide bond formation by three orders of magnitude in a well-defined in vitro system. These observations suggest that alkylative stress is likely to stall translation in vivo and necessitates the activation of ribosome-rescue pathways. Indeed, the addition of alkylation agents was found to robustly activate the transfer-messenger RNA system, even when transcription was inhibited. Our findings suggest that bacteria carefully monitor the chemical integrity of their mRNA and they evolved rescue pathways to cope with its effect on translation.

## Introduction

Nucleic acids are consistently experiencing damage from numerous endogenous and exogenous insults, including reactive-oxygen species, ultraviolet radiation, and alkylating agents (*Wurtmann and Wolin, 2009*; *Simms and Zaher, 2016*; *Yan and Zaher, 2019a*). In particular, the oxygen and nitrogen atoms of nucleobases are readily modified by alkylating agents. RNA is more susceptible to chemical insults than DNA, in part due to its exposed Watson-Crick (WC) hydrogen-bonding interface (*Hofer et al., 2005*; *Wyatt and Pittman, 2006*). Notably, the integrity of the WC face is paramount during codon recognition of the tRNA selection process, during which the three nucleotides of the codon base pair with those of the anticodon of the tRNA (*Zaher and Green, 2009a*). Changes that disrupt the ability of the mRNA to properly base pair with the cognate tRNA are then highly likely to reduce translational speed and fidelity (*Simms et al., 2014*; *Hudson and Zaher, 2015*; *Thomas et al., 2019*; *Hoernes et al., 2018*; *Hoernes et al., 2016*; *You et al., 2017*). To this end, several alkylative damage adducts have either been predicted or shown to be detrimental to the decoding process (*Hudson and Zaher, 2015*; *Hoernes et al., 2016*; *You et al., 2017*). For example, our group has previously shown that O6-methylguanosine ($m^6G$), which is highly mutagenic during DNA replication, interferes with the speed and accuracy of decoding when present in mRNA. Interestingly, the effect of $m^6G$ on translation was found to depend on its position within the codon (*Hudson and Zaher, 2015*). These observations suggest that given the nature of the decoding process, during which three nucleotides are read simultaneously, modifications to the mRNA can have complex effects on tRNA selection that cannot be solely predicted by their effect on base pairing.

N1-methyladenosine ($m^1A$) is an interesting modification because it has been the focus of several recent studies as a potential regulatory modification on mRNA (*Zhang and Jia, 2018*). However, a functional role for $m^1A$ in RNA metabolism is not without controversy. Depending on the $m^1A$-seq technique used, $m^1A$ has been found on as much as 20% of the transcriptome (*Dominissini et al.,*

\*For correspondence:
hzaher@wustl.edu

**Competing interests:** The authors declare that no competing interests exist.

2016; Li et al., 2016), and as low as nine sites only (Safra et al., 2017). Still, regardless of the method used to map the modification, over half of the identified m$^1$A adducts have been mapped to the coding region of transcripts, suggesting that modification is likely to affect ribosome function (Dominissini et al., 2016; Li et al., 2016). Notably, a specific regulatory role of this modification during translation has not been convincingly identified. On the contrary, studies generally support the hypothesis that m$^1$A exists primarily as a damage adduct that disrupts the decoding process (You et al., 2017). This idea is supported by studies conducted using E. coli translation extracts, for which the presence of m$^1$A at any of the three positions within the codon significantly decreased protein-synthesis yield (You et al., 2017). This effect on translation is not unexpected, considering the structure of m$^1$A; the addition of the methyl group to the N1 atom changes the hydrogen-bond donation and acceptance of the nucleobase. Furthermore, the modification introduces a resonance structure with a positive charge to the nucleobase. Consistent with these ideas, the presence of m$^1$A has been shown to cause local duplex melting in RNA (Zhou et al., 2016), and hence would likely occur during codon recognition of tRNA selection.

No known methyltransferase that specifically adds m$^1$A to mRNA has been identified. Instead, m$^1$A and several other adducts have been hypothesized to primarily result from reactions between RNA and endogenous and exogenous chemicals (Yan and Zaher, 2019a). In yeast, for example, the addition of alkylating compounds was found to significantly increase the levels of m$^1$A within mRNA, among several other adducts (Yan et al., 2019b). Interestingly, these compounds were also observed to activate eukaryotic-quality-control pathways known to be responsible for ribosome rescue, suggesting that alkylation stress causes ribosome stalling presumably due to its damaging effect on mRNA. These quality-control processes include the mRNA-surveillance pathway of no-go decay (NGD) and ribosome-quality-control (RQC) pathway responsible for the degradation of the associated incomplete nascent peptide (Simms et al., 2017a). Briefly, in eukaryotes, ribosome stalling results in ribosome pileup and eventual collisions. Collided ribosomes are recognized by an E3 ligase (Hel2 in yeast and ZNF598 in mammals), which adds K63-linked ubiquitin chains to ribosomal proteins (Matsuo et al., 2017; Simms et al., 2017b; Saito et al., 2015; Yan and Zaher, 2019c; Ikeuchi et al., 2019; Juszkiewicz and Hegde, 2017; Sundaramoorthy et al., 2017; Juszkiewicz et al., 2018). The ubiquitination is used as a signal to recruit downstream factors involved in mRNA degradation and ribosome splitting (Matsuo et al., 2017; Ikeuchi et al., 2019). The dissociated large-ribosome subunit, still bound to the peptidyl tRNA, is recognized by another E3 ligase (Ltn1 in yeast and listerin in mammals), which adds K48-linked ubiquitin to the nascent peptide, acting as a signal for its degradation by the proteasome (Bengtson and Joazeiro, 2010; Defenouillère et al., 2013; Brandman et al., 2012; Lyumkis et al., 2013).

Bacteria appear to have evolved entirely distinct mechanisms to deal with stalled ribosomes. This distinction between the two domains of life has been hypothesized to be the result of the divergent mechanisms utilized to terminate protein synthesis and recycle the ribosome (Simms et al., 2017a). In bacteria, at least four discrete rescue mechanisms have been identified. These include trans translation by the tmRNA as well as several other mechanisms that recruit alternative rescue factors (Arf proteins), which alone or in complex with release factors terminate protein synthesis in the absence of stop codons (Gagnon et al., 2012). Of these, the tmRNA system appears to be the most widely utilized and conserved rescue pathway (Keiler, 2015). tmRNA or transfer-messenger RNA is a unique molecule in that it contains a transfer RNA segment, which is aminoacylated by Ala-tRNA synthetase; and a messenger mRNA segment, which is used to switch the mRNA template from the defective mRNA to the molecule itself, and hence the name trans translation (Keiler et al., 1996; Moore and Sauer, 2007). Upon ribosome stalling, tmRNA binds the A site of the ribosome in a quaternary complex with elongation factor Tu (EFTu), its partner protein SmpB and GTP (Valle et al., 2003). Following peptidyl transfer to tmRNA, the original defective mRNA exits the ribosome, which then begins trans translation whereby the ribosome switches template to the mRNA part of tmRNA (Moore and Sauer, 2007; Gottesman et al., 1998). This mRNA encodes a peptide-degradation signal followed by a stop codon, ensuring that the incomplete peptide is degraded by the ClpXP protease system following canonical termination and recycling of the ribosome (Keiler and Sauer, 1996).

We note that the tmRNA system has been extensively studied in the context of truncated mRNAs, which result from a myriad of conditions including ribosome stalling (Moore and Sauer, 2007), but whether the process is also activated in response to chemical insults that modify RNA is unknown. Here we explored the impact of alkylation damage on translation and investigated its effect on

tmRNA activity in *E. coli*. We first demonstrated that treating *E. coli* with common alkylating agents increases the levels of several potentially disruptive alkylative adducts, including m$^1$A. To quantify the effect of m$^1$A on decoding, we used a well-defined in vitro translation system to measure observed rates of peptide bond formation in the absence and presence of the modification in mRNA. The modification reduced the rate of peptide bond formation by almost three orders of magnitude. The decrease in peptide bond formation was also accompanied by a reduction in the endpoint of the peptidyl-transfer reaction, suggesting that the modification affects the proofreading phase of tRNA selection. Highlighting the disruptive effect of m$^1$A on tRNA selection was the observation that aminoglycosides, which increase peptide bond formation on mismatched-codon-anticodon interactions, had no effect on reactions with m$^1$A-programmed mRNAs. These findings suggest that alkylation stress, which increases the levels of disruptive adducts such as m$^1$A, causes widespread stalling in bacterial cells and activates rescue pathways. In complete agreement with these ideas, we documented robust activation of tmRNA in *E. coli* cells treated with alkylating agents, even when transcription was inhibited. Interestingly, DNA damage was also found to trigger tmRNA activation, but this activation was significantly suppressed when transcription was inhibited. Hence, alkylation stress is likely to significantly impact translation through its effect on mRNA and rescue pathways such as *trans* translation are responsible for dealing with its consequences. Consistent with these proposals, *E. coli* strains lacking functional tmRNA were found to be sensitive to alkylating agents and exhibited delayed recovery compared to wild-type (WT) cells. Collectively our data suggest chemical damage to mRNA is highly detrimental to cellular homeostasis, even in organisms where mRNA is highly transient.

## Results

### Treatment of *E. coli* with MMS or MNNG causes significant increases in alkylative damage of RNA

As an initial step to explore the impact of alkylative damage on translation in bacteria, we sought to establish methods that allowed us to robustly induce RNA alkylation in *E. coli*. We have recently shown that the addition of methyl methanesulfonate (MMS) to growing yeast cultures leads to the accumulation of alkylation adducts in RNA (*Yan et al., 2019b*). As a result, we expected the compound to similarly modify RNA in bacteria. To ensure that any effect we observe on translation is a general response to RNA alkylation, we also chose to study the impact of methylnitronitrosoguanidine (MNNG) on RNA metabolism and translation. MMS and MNNG work through different nucleophilic-substitution mechanisms to alkylate nucleic acids. MMS alkylates its target through an SN$_2$-type mechanism, while MNNG reacts through an SN$_1$-type one (*Wyatt and Pittman, 2006*); therefore, we expected to observe differences in the types and levels of adduct that each agent generated. To identify and determine the abundance of each modification, total RNA was isolated from mid-log growing *E. coli* cells that had been mock treated (DMSO) or treated with MMS (0.1%) or MNNG (5 µg/mL) for 20 min. RNA was consequently digested to nucleotide monophosphates by incubating it with P1 nuclease. Nucleosides were generated by incubating the P1-reaction products with calf-intestinal phosphatase (CIP); and analyzed by liquid chromatography – mass spectrometry (LC-MS). We generated standard curves for each of the unmodified nucleosides, as well as for N1-methyladenosine (m$^1$A), N6-methyladenosine (m$^6$A), N1-methylguanosine (m$^1$G), O6-methylguanosine (m$^6$G), and N3-methylcytidine (m$^3$C) in order to directly quantify the modified nucleosides within each treatment (*Figure 1—figure supplement 1*).

To confirm whether our LC-MS methods are relatively accurate, we measured the ratio of m$^6$A/A levels in untreated cells, which has been estimated to be 0.3% in *E. coli* (*Deng et al., 2015*). In agreement with these earlier studies, we measure a ratio of 0.6% (*Figure 1A*). Also as expected, we did not detect an increase in m$^6$A levels after treatment with either alkylating agent (*Figure 1A*), as neither MMS nor MNNG alkylate the N6 position of adenosine (*Wyatt and Pittman, 2006*). Additionally, MMS and MNNG have been shown to react with N1 of G, albeit it with reduced efficiency (*Wyatt and Pittman, 2006*); but we did not observe significant increases in m$^1$G levels (*Figure 1B*). These observations can be rationalized by the fact that m$^1$G is a natural modification of *E. coli* tRNA and rRNA (*Machnicka et al., 2013*), which significantly increases the background levels of the modification. Indeed, the base level of m$^1$G is at least 200-fold higher than that of m$^1$A (*Figure 1*).

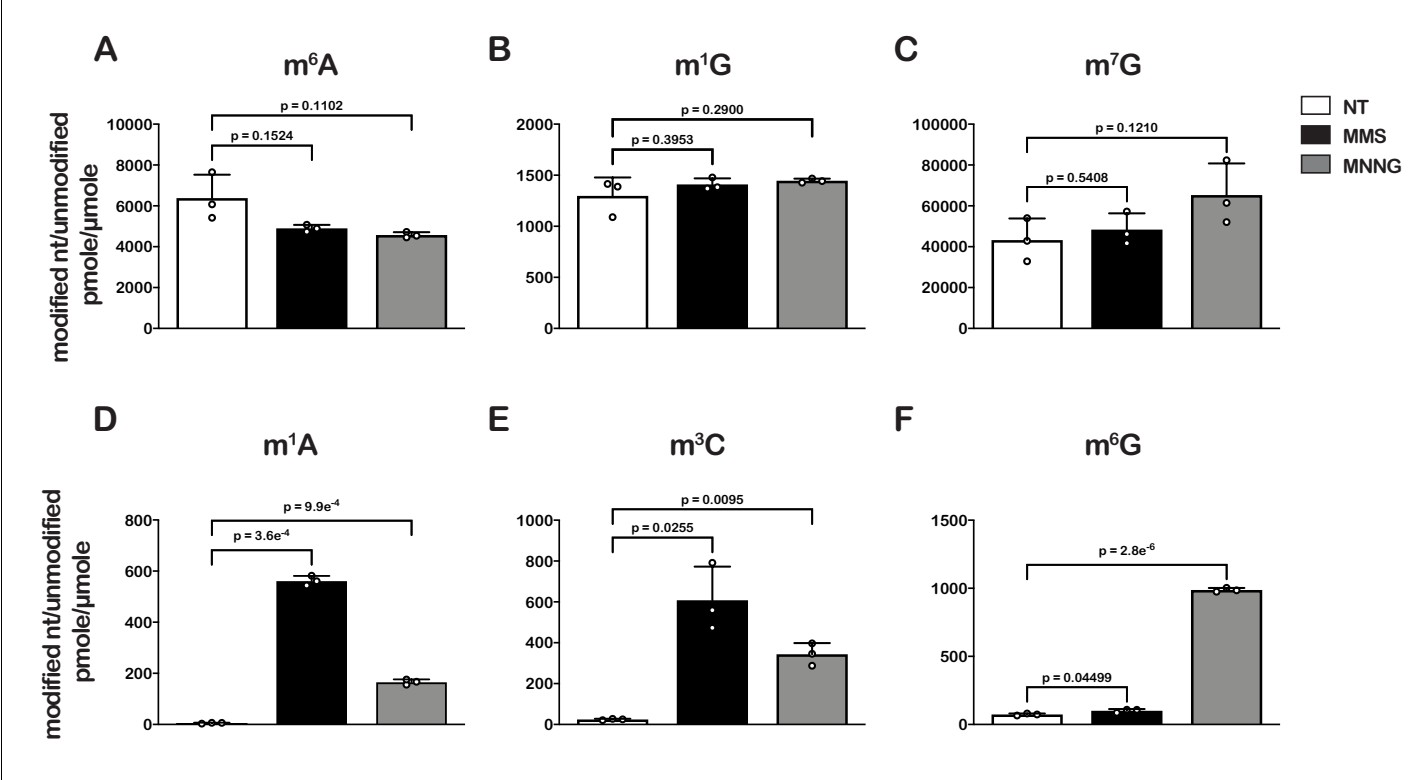

**Figure 1.** Treatment of *E. coli* with MMS and MNNG results in significant accumulation of alkylative-damage adducts in RNA. (A–E) Bar graphs showing the amount of the indicated modified nucleotides relative to their unmodified parent in untreated (white bars), MMS-treated (black bars), and MNNG-treated (gray bars) cells. The values plotted are the averages of three biological repeats and the error bars represent standard deviations around the mean. Significant differences in mean, as denoted by p<0.05, were determined by Welch's t-test.

The online version of this article includes the following source data and figure supplement(s) for figure 1:

**Source data 1.** Quantification of alkylative-damage adducts.
**Figure supplement 1.** LC-MS calibration curves for modified and unmodified nucleosides.
**Figure supplement 1—source data 1.** LC-MS calibration curve source data.

Similarly, since $m^7G$ is a natural modification in rRNA and tRNA (*Machnicka et al., 2013*), we observe no significant change to its levels upon MMS and MNNG additions (*Figure 1C*).

Contrary to $m^6A$ and $m^1G$, we measured 10- to 200-fold increases in $m^3C$ and $m^1A$ relative levels in cells treated with MMS or MNNG (*Figure 1D and E*), and a more than tenfold increase in $m^6G$ relative levels but only in those treated with MNNG (*Figure 1F*). These findings are consistent with previous studies showing that the O6 position of guanosine is reactive with MNNG but not MMS and that $m^3C$ and $m^1A$ are minor alkylative adducts in double-stranded DNA (*Beranek, 1990*) but are substantially more reactive as nucleophiles in the absence of hydrogen bonding (*Bodell and Singer, 1979*). This same increase in reactivity for N1 to G in single-stranded RNA is not observed because it is a secondary amine with an adjacent carbonyl group which is less reactive than N1 of A and N3 of C, both of which have the higher reactivity profiles of amidine groups (*Wyatt and Pittman, 2006*).

## N1-methyladenosine has drastic effects on peptide bond formation in vitro

Having established a method of increasing the levels of alkylative adducts in *E. coli* mRNA, we next became interested in assessing the effects of some of these modifications on translation using a well-defined in vitro system (*Zaher and Green, 2009b*). In particular, we focused on modifications whose levels significantly increase in the presence of alkylation stress and have been documented to disrupt the base-pairing properties of the nucleobase. Therefore, $m^1A$ and $m^3C$ were obvious

candidates as both satisfied these two requirements. Since m[3]C phosphoramidite was not commercially available, we opted to study the effects of m[1]A on translation in our system. Briefly, we generated ribosomal initiation complexes carrying f-[35S]-Met-tRNA[fMet] in the P site and displaying either an unmodified GAA codon or an m[1]A adduct at the second position (G[m1]AA) in the A site (*Figure 2B*). The GAA codon is decoded by its cognate Glu-tRNA[Glu].

Both complexes were reacted with Glu-tRNA[Glu]•EFTu•GTP ternary complex, and the formation of f-[35S]-Met-Glu dipeptide was followed as a function of time by taking aliquots at the appropriate times and quenching the reaction with potassium hydroxide, which hydrolyzes the peptide away from the tRNA. The resulting dipeptide was resolved from unreacted fMet using an electrophoretic thin-layer chromatography (TLC) system (*Youngman et al., 2004*). The relative amount of dipeptide was quantified using phosphorimaging and fitted to a single-exponential to determine the observed rate of peptide bond formation. As expected, m[1]A was found to severely inhibit peptide bond formation. The observed rate of $0.15 \text{ s}^{-1}$ for the G[m1]AA complex was measured to be more than 250-fold slower relative to the observed rate of $58 \text{ s}^{-1}$ for the unmodified GAA complex. Furthermore, the endpoint of the m[1]A reaction, which reports on the efficiency of proofreading (*Pape et al., 1999*), was found to be ~tenfold lower relative to the unmodified one (*Figure 2C*). These results demonstrate that m[1]A is highly detrimental to the tRNA selection process and is highly likely to stall ribosomes in vivo.

Next, we wondered whether the m[1]A:U base pair between the codon and the anticodon of the tRNA is recognized by the ribosome as a simple mismatch or is more disruptive than that. During tRNA selection, aa-tRNAs that have only one mismatch between their anticodon and the A-site codon are termed near-cognate (*Gromadski and Rodnina, 2004a*). Although the ribosome efficiently discriminates against the incorporation of near-cognate aa-tRNAs, the addition of certain aminoglycosides alters the selection process and allows for their misincorporation (*Pape et al., 2000*). For instance, paromomycin binds the decoding center of the ribosome, which leads to local conformational changes that lower the energy barrier required to shift the ribosome into a 'closed' conformation (*Fourmy et al., 1996*; *Ogle et al., 2002*). This conformation typically occurs in the presence of cognate aa-tRNAs and is a prerequisite for their accommodation into the active site of the ribosome for peptidyl transfer to take place (*Ogle et al., 2002*). By contrast, the addition of aminoglycosides to reactions containing non-cognate aa-tRNAs, which harbor more than one mismatch between their anticodon and the A-site codon, has little to no effect on peptide bond formation (*Pape et al., 2000*; *Gromadski and Rodnina, 2004b*). As a result, the effect of aminoglycosides on peptidyl transfer can be used as a diagnostic for the sort of interactions taking place in the decoding center and how modifications alter them. For instance, we recently showed that the addition of paromomycin and streptomycin de-represses the effects of the oxidative adduct 8-oxoguanosine (8-oxoG) on peptide bond formation (*Thomas et al., 2019*). These findings suggest that the ribosome recognizes this modification as a simple mismatch when paired with its partner nucleotide. Interestingly and in contrast to 8-oxoG, in the presence of m[1]A, we found the addition of paromomycin to have little to no effect on the observed rate of peptide bond formation. In particular, we measured observed rates of $0.15 \text{ s}^{-1}$ and $0.2 \text{ s}^{-1}$ in the absence and presence of an antibiotic, respectively (*Figure 3*). Additionally, the endpoints of the very same reactions were 0.10 and 0.11, respectively. These findings suggest that not only is m[1]A unable to base pair with U in the decoding center but that the modification significantly distorts the codon-anticodon helix. This distortion results in the aa-tRNA, that is otherwise cognate, to be recognized as a non-cognate one by the ribosome.

To provide further insights into how an m[1]A modification within the codon changes the base-pairing properties of the nucleotide, we investigated how it alters miscoding in the presence of all near- and non-cognate tRNAs. We performed an aa-tRNA-reactivity survey, in which we reacted the previously described unmodified- and m[1]A-containing-initiation complexes (*Figure 2*) with all 20 possible aa-tRNA isoacceptors for 2 min (*Figure 4*). As anticipated, for the codon containing the unmodified adenosine, we observed no significant dipeptide accumulation except in the presence of the cognate Glu-tRNA[Glu] ternary complex (*Figure 4*). For the codon containing m[1]A, no significant dipeptide accumulation occurred in the presence of any aa-tRNA. These findings strongly suggest that the modification inhibits base pairing altogether and does not alter the base-pairing preference of the nucleotide.

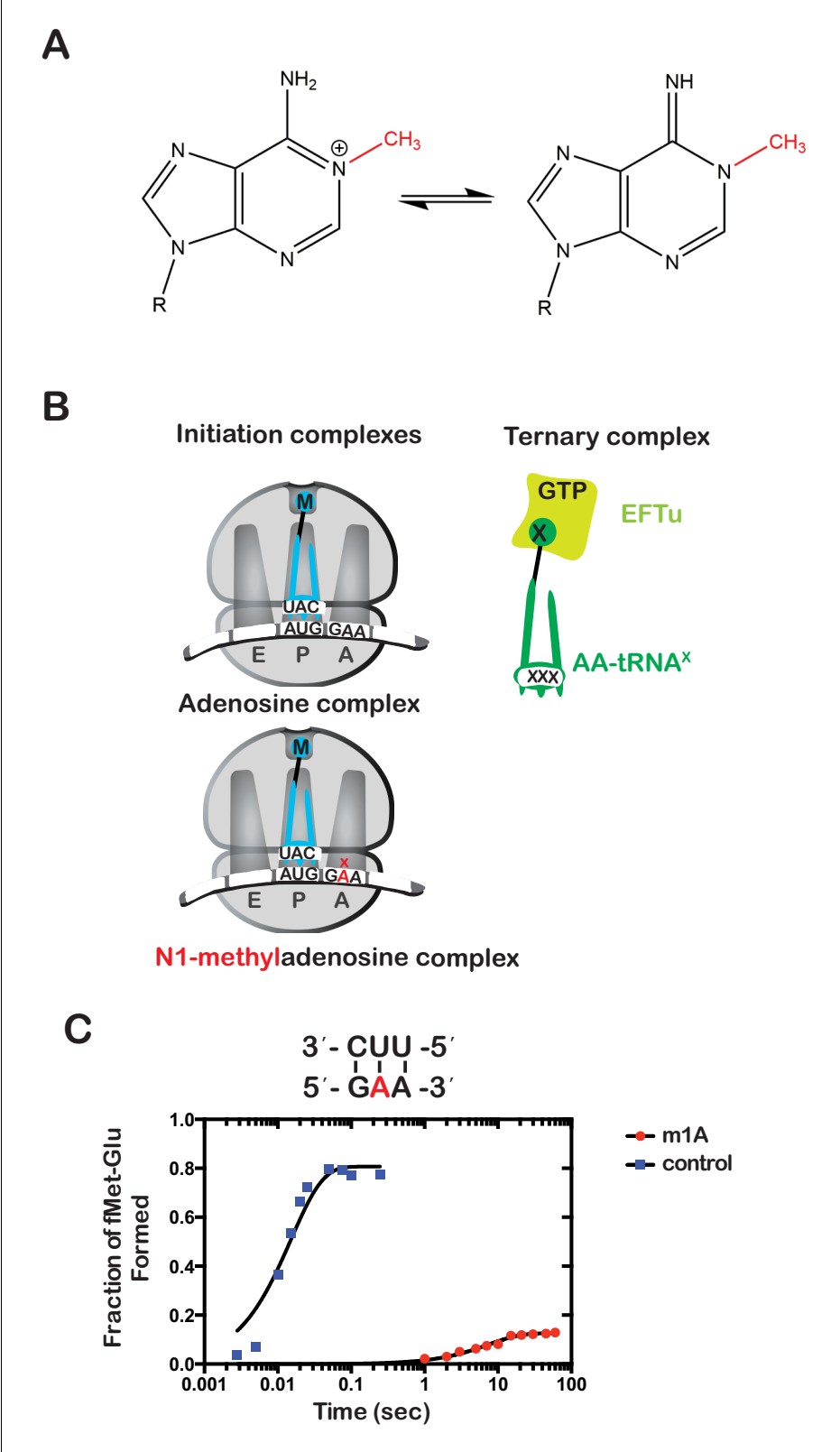

**Figure 2.** N(1)-methyladenosine (m$^1$A) in mRNA significantly decreases the rate and endpoint of peptide bond formation in vitro. (**A**) Chemical structure of m$^1$A. The N1-methyl group is highlighted in red, and the resonance structure of the molecule is represented. (**B**) Schematic representation of adenosine and m$^1$A initiation complexes encoding for the dipeptide Met-Glu. Both complexes contain the initiator fMet-tRNA$^{fMet}$ in the P site; the A complex displays a GAA codon, while the m$^1$A complex displays a G$^{m1}$AA codon in the A site. (**C**) Representative time-courses of peptide bond-

*Figure 2 continued on next page*

*Figure 2 continued*

formation reactions between initiation complexes programmed with unmodified mRNA (blue) or an m1A-modified one (red) m1A, and Glu-tRNA$^{Glu}$ ternary complex.

The online version of this article includes the following source data for figure 2:

**Source data 1.** m1A kinetics source data.

## Alkylative damage of RNA increases tmRNA activity in vivo

Our mass-spectrometry analysis indicated that the addition of MMS and MNNG to *E. coli* significantly increases the levels of several modified nucleotides, including m1A (*Figure 1*), which we showed to have drastic effects on the peptide bond formation (*Figure 2*). As a result, we predicted the addition of these compounds to stall translation in vivo and to activate rescue pathways such as *trans* translation by tmRNA. As mentioned earlier, tmRNA encodes a peptide-degradation sequence, which ensures incomplete nascent peptides are c-terminally tagged for rapid degradation by cellular proteases. The mRNA sequence encoding this tag can be modified without significantly affecting the *trans* translation activity of tmRNA. In particular, substituting the sequence for one that encodes a His-tag has been successfully used to identify tmRNA targets (*Roche and Sauer, 2001*). We took advantage of this tmRNA variant to assess its relative activity in response to alkylation stress by probing with the anti-His antibody. Consistent with our proposal that alkylation stress activates the tmRNA system, we observed two- to threefold increases in His$_6$-tagging levels upon the addition of MMS and MNNG to *E. coli* expressing tmRNA-His$_6$ (*Figure 5A and B*). As expected, this increase in tmRNA activity was accompanied by activation of the adaptive response as judged by the accumulation of Ada protein, the principal mediator of the process in response to alkylation damage (*Jeggo, 1979*; *Sakumi and Sekiguchi, 1989*; *Uphoff et al., 2016*). We also observe activation of the SOS response as evaluated by the accumulation of its principal mediator RecA (*Cox, 2007*) It is worth noting that at this concentration of MMS and duration of treatment, we observe little to no effect on cellular viability as assessed by spot assays (*Figure 5—figure*

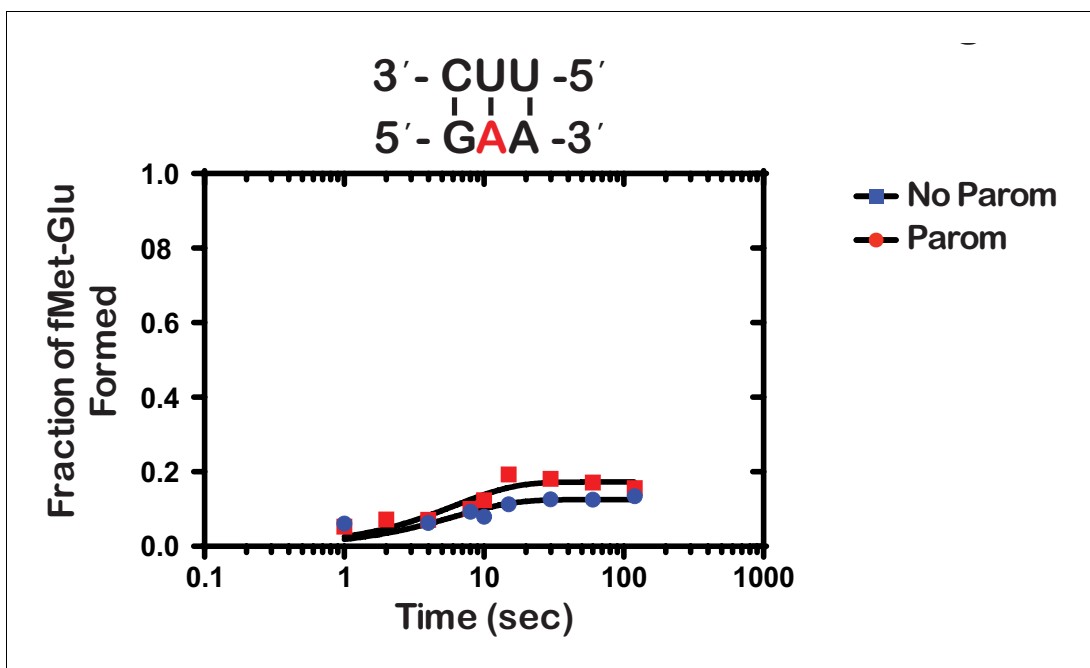

**Figure 3.** Paromomycin does not rescue the effect of m1A on peptide bond formation. Representative time-courses of peptide bond-formation reactions between initiation complexes programmed with m1A mRNA and Glu-tRNA$^{Glu}$ ternary complexes in the absence (blue) and presence (red) of paromomycin.

The online version of this article includes the following source data for figure 3:

**Source data 1.** Paromomycin kinetics source data.

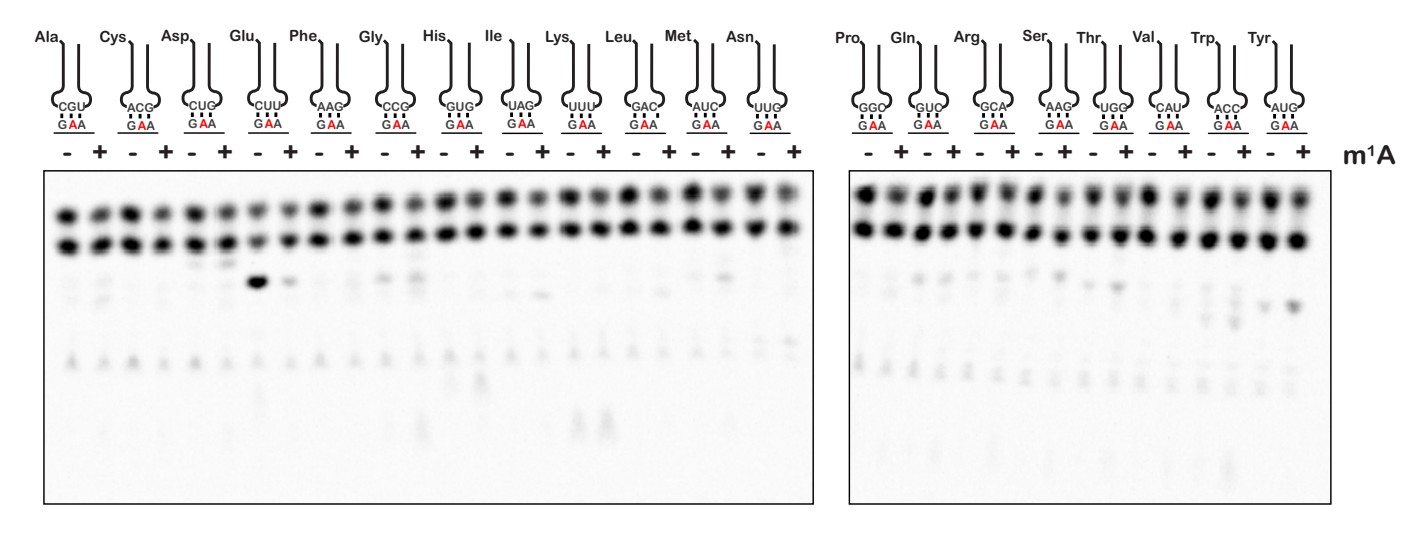

**Figure 4.** m$^1$A does not alter the reactivity of ribosomes with near-cognate and non-cognate aa-tRNA. Phosphorimager scan of electrophoretic TLCs used to follow dipeptide-formation reactions between unmodified and m$^1$A-modified complexes with all canonical aa-tRNA ternary complexes.

*supplement 1A and B*). Altogether our data suggest that alkylation stress stalls translation in bacteria and activates *trans* translation by tmRNA.

Under normal conditions, His-tagged products resulting from tmRNA activation are targeted for degradation by the ClpAP, ClpXP, and Lon proteases, presumably because they are incomplete peptides that are likely to misfold. Indeed, previous reports have shown the deletion of Δ*clpP*, *clpX*, and *lon* genes to result in increased accumulation of His-tagged products (*Moore and Sauer, 2005*). These observations suggest that the amount of His-tagging that we detect in the presence of MMS and MNNG is likely an underestimate of the tmRNA-activation levels. To obtain a more accurate measure of *trans* translation in response to alkylation stress, we treated *E. coli* null mutants of *clpP*, *clpX*, and *lon* with MMS and MNNG. As expected, the levels of His-tagging triggered by the addition of alkylation compounds increased by more than twofold in the absence of these proteases (*Figure 5—figure supplement 2*). In agreement with previous studies that suggested Ada is subject to proteolysis (*Sedgwick, 1989*), in *clpP*, *clpX*, and *lon* cells, the factor was observed to accumulate even in the absence of alkylation stress.

As has been noted in a previous study, the His-tagging patterns between each of the samples appear almost identical (*Moore and Sauer, 2005*). This was a surprising observation, as we expected the alkylative damage to be randomly located throughout the transcriptome, thereby resulting in a more uniform streak of his-tagging or banding patterns that varied from sample to sample. A plausible explanation for this banding pattern is that the antibody we used displays a preference for certain peptide sequences. We tested this idea by probing with antibodies from three different manufacturers and compared the observed pattern. Interestingly, each antibody displayed a unique pattern, with some recognizing only a small subset of potential peptides (*Figure 5—figure supplement 3*). Thus, the observed consistent His-tagging pattern is likely due to an artifact of the His antibodies rather than a particular subset of proteins that are preferentially His-tagged at certain locations.

The accumulation of His-tagged peptide products suggests that alkylative damage of RNA is stalling ribosomes in vivo and activating tmRNA. However, the alkylative damage from MMS and MNNG damages DNA as well as RNA (*Wyatt and Pittman, 2006*). This is supported by the increase in RecA that we observe upon treatment with alkylating agents, which is a protein that is essential for the maintenance and repair of DNA (*Cox, 1991*; *Figure 5A*). Therefore, it is also possible that the resulting increase in His-tagging was primarily due to the production of truncated transcripts that are produced by stalled RNA polymerase on damaged DNA. As most of these truncated transcripts lack a stop codon, they represent the classical targets of tmRNA (*Moore and Sauer, 2007*). To ensure that the observed His-tagging was due to RNA damage rather than truncated RNA

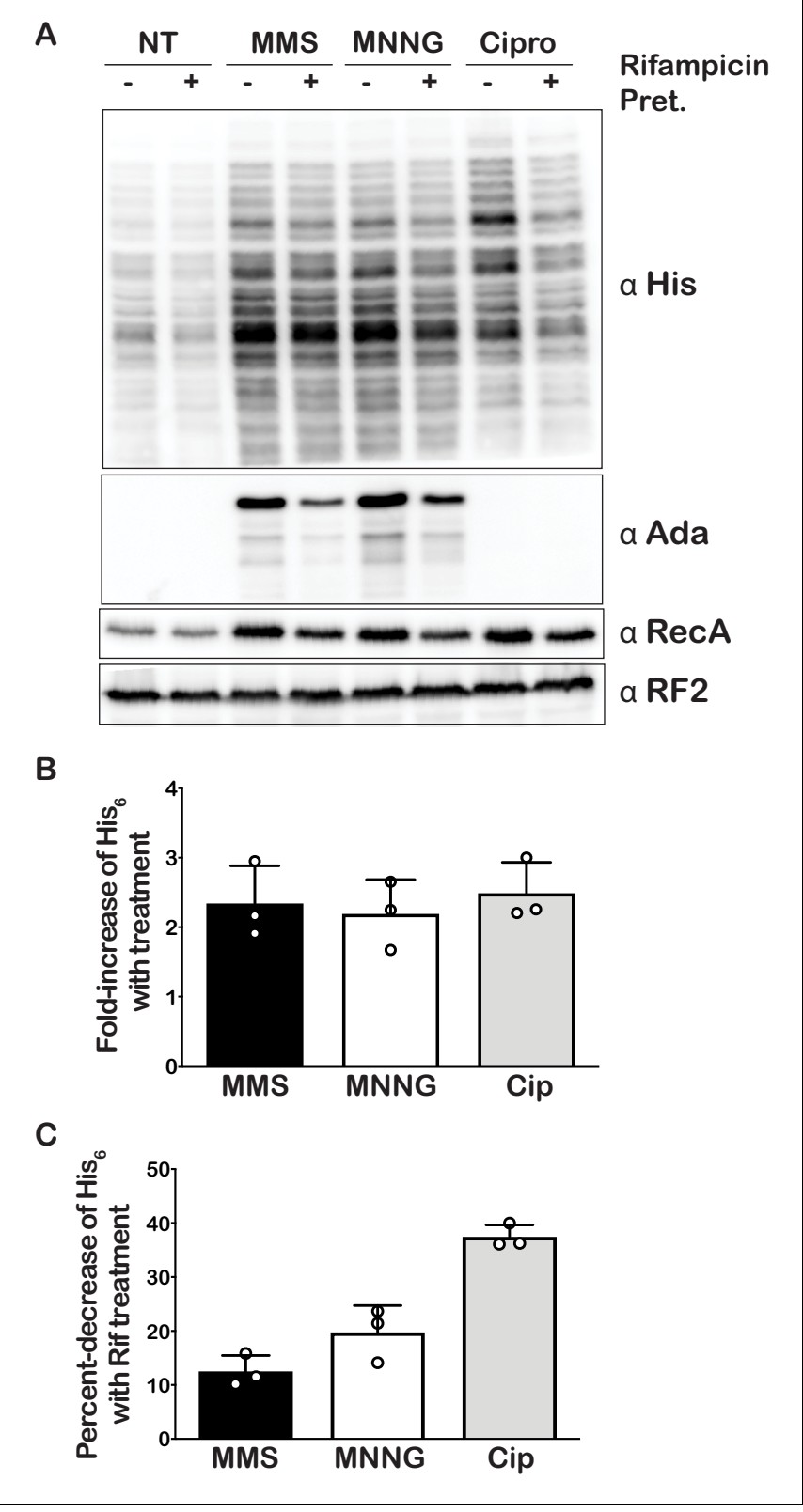

**Figure 5.** Alkylative stress activates *trans* translation in *E. coli* in a transcription-independent manner. (**A**) Western blot analyses of total protein isolated from an *E. coli* strain expressing tmRNA-His$_6$. Cells were either untreated or treated with MMS, MNNG, or ciprofloxacin. Additionally, for each condition, cells were either mock pretreated or received a rifampicin pretreatment. Blots were probed with the indicated antibodies. (**B**) Bar graph showing the relative change in His signal (tmRNA activity) as a result of the addition of each of the indicated compounds. (**C**) Bar graph used to depict

*Figure 5 continued on next page*

*Figure 5 continued*

the fold-decrease of His$_6$ levels upon pre-treatment with rifampicin for each of the indicated treatments. In all cases, the initial His signal was normalized to that corresponding to RF2 levels before it was used to calculate the relative change. Three independent experiments were used to obtain the bar graphs, with the mean values plotted and the error bars representing the standard deviation around the mean.

The online version of this article includes the following source data and figure supplement(s) for figure 5:

**Source data 1.** Quantification of tmRNA tagging.
**Figure supplement 1.** WT and Δ*ssrA E. coli* exhibit similar survival phenotypes after treatment with MMS.
**Figure supplement 1—source data 1.** Spot assay quantification.
**Figure supplement 2.** Deletion of ClpXP and Lon proteases results in further accumulation of tmRNA-induced His$_6$ tagging of peptides upon alkylative stress.
**Figure supplement 3.** Different His antibodies display unique banding patterns on western blots.
**Figure supplement 4.** Ciprofloxacin, but not mitomycin C, increases His$_6$ tagging by tmRNA.
**Figure supplement 5.** Optimal His$_6$ tagging and activation of Ada and RecA levels are achieved after 20 min of MMS treatment.
**Figure supplement 6.** Significant transcriptional runoff is achieved after 10 s of rifampicin treatment.

produced from damaged DNA, we pre-treated cells with rifampicin approximately 40 s before treating with damaging agents. Rifampicin is an inhibitor of RNA polymerase initiation; therefore, the pretreatment with rifampicin halts transcriptional initiation and allows us to separate the effects of DNA damage from RNA damage (*Wehrli, 1983*). In addition to MMS and MNNG, we also treated cells with ciprofloxacin and mitomycin C. Ciprofloxacin is an antibiotic that inhibits the ligation activity of DNA gyrase and topoisomerase IV but not the cleavage activity, thereby causing the topoisomerases to create double-stranded breaks in DNA (*LeBel, 1988*). Mitomycin C causes intra- and inter-strand DNA crosslinks that can block the activities of DNA polymerase and RNA polymerase (*Verweij and Pinedo, 1990*). We predicted that these two agents, which specifically cause DNA damage, would produce truncated transcripts and result in His-tagging by tmRNA. For these two drugs, however, the pretreatment with rifampicin is expected to inhibit the accumulation of His-tagged products. In agreement with our model, pretreatment with rifampicin only slightly decreased the amount of His-tagging induced by MMS and MNNG (10–20%) (*Figure 5C*), suggesting that the majority of the observed tmRNA activity is due to direct damage to the mRNA. In contrast, the same rifampicin pretreatment combined with ciprofloxacin treatment decreased the amount of His-tagging by almost twofold. Interestingly, for the mitomycin C treatment, we did not observe any His-tagging even though we could infer that significant DNA damage had occurred as assessed by the increase in RecA levels (*Figure 5—figure supplement 4*). Regardless of this last observation, we conclude that while truncated transcripts produced from damaged DNA do activate tmRNA, the predominant trigger of tmRNA activation in MMS- and MNNG-treated samples is alkylative damage of RNA.

## Addition of MMS-treated mRNA to *E. coli* extracts results in trans translation

To add further support for our model that mRNA modification by alkylation agents activates tmRNA, we prepared S30 extracts from the protease-deficient *E. coli* strain that harbors His-tagged tmRNA and assessed the translation of chemically damaged mRNAs in a cell-free system. We amplified the *prmc* gene, added a FLAG tag to its N-terminus, and used it as a template for in vitro transcription (*Figure 6A*). The resulting mRNA was either mock treated or treated with 0.5% MMS, which did not alter its integrity as assessed by denaturing agarose electrophoresis (*Figure 6B*). $^{35}$S labeling and western-blotting analysis (using anti-FLAG antibody) revealed that both mRNA species were translated efficiently in our extract, with the MMS-treated mRNA yielding slightly less full-length product. These observations suggest that modification by MMS results in stalling on this mRNA (*Figure 6C*). Notably, probing with anti-His antibody revealed that tmRNA is significantly activated in reactions containing damaged mRNA (*Figure 6C*). The observation that tmRNA tagging occurs on modified mRNA in a system where transcription is not occurring provides important support for our model that *trans* translation responds to ribosome stalling on damaged mRNA.

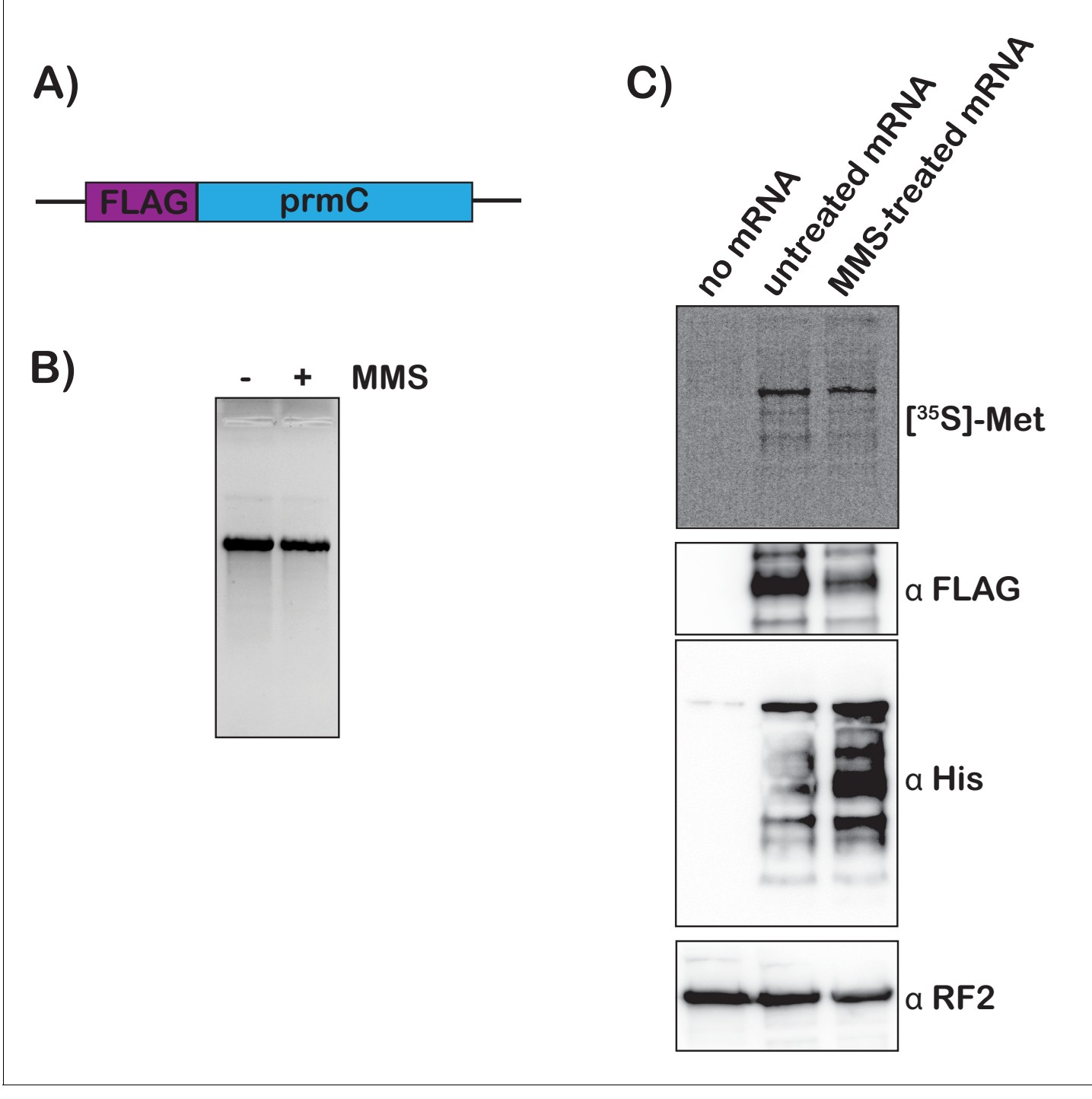

**Figure 6.** Translation of MMS-treated mRNA results in tmRNA tagging in an S30 extract.  (**A**) Schematic of the mRNA used in the in vitro translation assays. (**B**) Fluorescence image of EtBr-stained gel was used to visualize the mock-treated and MMS-treated RNAs. (**C**) Top shows a phosphor-imager scan of a PVDF-transfer membrane of a bis-tricine gel that was used to separate products from the in-vitro-translation reactions containing the indicated mRNAs. Bottom is western-blotting analyses of the same membranes with the indicated antibodies. Shown is a representative of two assays. The online version of this article includes the following figure supplement(s) for figure 6:

**Figure supplement 1.** Deletion of *ssrA* gene results in stabilization of m$^1$A-modified mRNA.

## Deletion of the *ssrA* gene leads to stabilization of m$^1$A-modified mRNAs

Previous studies indicated that in addition to *trans* translation, tmRNA plays a role in the degradation of its target mRNAs. In particular, the molecule facilitates the recruitment of RNase R, which initiates nonstop mRNA decay (*Ge et al., 2010*). To this end, deletion of the *ssrA* gene results in the accumulation of nonstop mRNAs. As our findings suggest that ribosomes stalled on modified mRNAs engage the *trans*-translation machinery, we hypothesized that tmRNA is also likely to play a role in the degradation of chemically damaged mRNAs. We note that analogous studies by our group conducted in yeast took advantage of the polyA tail of mRNAs in eukaryotes in order to purify them before modification analysis by LC-MS (*Yan et al., 2019b*). Unfortunately, owing to the lack of a similar feature in bacterial mRNAs, a similar strategy for *E. coli* cannot be used. Nonetheless, we were able to use an indirect approach to qualitatively assess the levels of m$^1$A modification in mRNAs. In this approach, total RNA is resolved on denaturing agarose gels, transferred to nitrocellulose, and probed with antibodies specific to the nucleoside of interest (*Meyer et al., 2012*). In our hands, antibodies raised against m$^1$A did not produce a signal even with positive standards. By contrast, we were able to obtain a robust signal with an anti m$^6$A antibody (*Figure 6—figure supplement 1A*). Since m$^1$A can readily isomerize to m$^6$A through Dimroth rearrangement by base treatment (*Macon and Wolfenden, 1968*), immunoblotting of base-treated RNAs with this antibody can be used to assess the levels of m$^1$A. As expected, and in agreement with our LC-MS analysis (*Figure 1*), without base treatment immunoblotting with m$^6$A antibody showed that MMS addition to cells leads to a slight decrease in the signal, suggesting that the treatment does not alter the levels of this modification significantly. By contrast, base treatment of the gel before transfer revealed that treating cells with MMS causes a significant increase in the signal for RNA species that co-migrated between rRNA and tRNA species, which we attribute to mRNA species having m$^1$A modification (*Figure 6—figure supplement 1B*). More important was the observation that the MMS-induced signal was much higher in the Δ*ssrA* cells (*Figure 6—figure supplement 1B*). Our observations suggest that not only does tmRNA rescue ribosomes stalled on modified mRNAs, but that m$^1$A levels in mRNA increase in its absence.

## The ability to rescue stalled ribosomes is important for cellular recovery after alkylative damage

Even though the ssrA gene that encodes tmRNA is highly conserved in bacteria, previous studies have shown that ΔssrA *E. coli* strains show no appreciable growth phenotype under standard laboratory conditions but do exhibit delayed growth under certain stress conditions (*Nishikawa et al., 2006*; *Karzai et al., 1999*). Since we observed that tmRNA is utilized to rescue ribosomes stalled due to damaged RNA, we hypothesized that the ability of cells to rescue stalled ribosomes is important for cellular recovery upon treatment with alkylating agents. To test this, we treated Δ*ssrA* and WT cells with either 0.5% MMS or 20 µg/mL MNNG, washed the cells to remove the alkylating agent, and allowed them to recover while monitoring growth. As expected, in the absence of any pretreatment, Δ*ssrA* and WT cells recover at approximately the same rate. However, after treatment with MMS or MNNG, Δ*ssrA* cells have an approximately 1.5 hr lag in their recovery compared to WT cells (*Figure 7*). To rule out cellular death as a cause for the observed lag, we treated both Δ*ssrA* and WT cells with 0.5% MMS for 20, 40, and 60 min, washed the cells to remove the alkylating agents, and performed spot assays to quantify cell death. We observe similar levels of cell death for Δ*ssrA* and WT cells, suggesting that the observed lag in recovery time is not due to a difference in the number of cells killed, but rather the ability of the cells to quickly recover after treatment with alkylating agents (*Figure 5—figure supplement 1C and D*). Collectively, our findings suggest that cells employ *trans* translation to rescue ribosomes stalled on damaged mRNA to quickly recover from alkylation stress.

## Discussion

Several recent reports have shown that mRNA can be enzymatically modified to regulate its function (*Peer et al., 2017*). Among these modifications, m$^6$A is arguably one of the most significant ones and appears to play important roles in the regulation of gene expression (*Wang et al., 2014*).

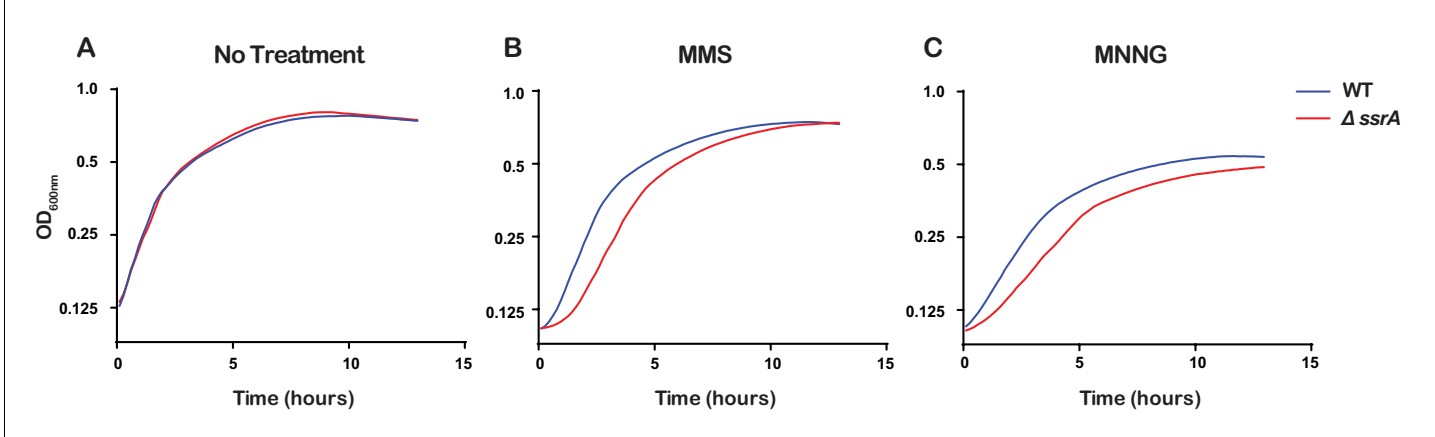

**Figure 7.** Ribosome rescue by tmRNA is important for cellular recovery after treatment with alkylating agents. (**A–C**) Growth curves, as measured by change in $OD_{600nm}$ as a function of time, for the indicated cells following a treatment with the denoted compound. Average of three replicate growth assays is plotted.

The online version of this article includes the following source data for figure 7:

**Source data 1.** Growth curve source data.

Pseudouridine ($\Psi$), due to its high abundance in mRNA, has recently emerged as another potentially important modification. Interestingly, pseudouridine-modified (and derivatives of $\Psi$) mRNAs have been shown to be effective for biotechnological purposes as they are less immunogenic and produce significantly higher levels of functional proteins than their unmodified counterpart (*Karikó et al., 2008*). Although much has been revealed about the role the so-called intentional modifications serve in regulating various aspects of mRNA metabolism, most chemical modifications of mRNA are disruptive damage adducts (*Simms and Zaher, 2016*). Previous work had shown that several alkylative damage adducts, including $m^1A$ and $m^6G$, drastically slow translation and increase miscoding in vitro (*Hudson and Zaher, 2015*; *You et al., 2017*). However, little is known about how alkylative damage of mRNA elicits cellular responses in vivo, and especially in bacteria. Additionally, the quantitative effects of $m^1A$ on the speed and accuracy of translation are not fully understood. Here we used compounds that alkylate nucleic acids to introduce damage adducts to bacterial RNA and assessed cellular responses to potential defects to translation. A priori, we hypothesized that the main ribosome rescue system in bacteria, the *trans*-translation pathway, works to release ribosomes stalled on damaged mRNA. Indeed, we find that upon treatment of *E. coli* with alkylating agents, *trans*-translation activity significantly increased (*Figure 5*). Furthermore, we show that cells lacking tmRNA display prolonged recovery from alkylation stress relative to wild-type cells (*Figure 7*).

This response to alkylation-mediated changes to translation is likely to be general, as it was nearly identical regardless of the specific compound used to elicit the damage. We used two alkylating agents, MMS and MNNG, which work through an $SN_2$- and $SN_1$-type nucleophilic substitution mechanism, respectively (*Wyatt and Pittman, 2006*). Indeed, LC-MS analysis revealed that the addition of the compounds results in different signatures of RNA modification. For instance, the addition of MNNG leads to a more than tenfold increase in $m^6G$ levels, whereas that of MMS leads to a modest <1.5-fold increase for the same modification (*Figure 1*). This is consistent with studies showing that MNNG produces a greater percentage of O-methyl adducts (*Wyatt and Pittman, 2006*).

Our group had previously utilized an in vitro reconstituted bacterial translation system to investigate the impact of $m^6G$ on decoding (*Hudson and Zaher, 2015*); however, we had not analyzed the effects of $m^1A$. We hypothesized that the positively charged resonance structure of $m^1A$ would disrupt its ability to base pair. During decoding on the ribosome, the first and second position of the codon-anticodon helix are closely monitored by rRNA nucleotides as well as ribosomal-protein amino acids, ensuring that only Watson-Crick base pairs are accepted (*Ogle et al., 2002*; *Wimberly et al., 2000*; *Carter et al., 2001*). We chose to analyze $m^1A$ in the second position of the codon rather than the first because we wanted to avoid possibly altering the interaction between

the initiator tRNA and the P-site codon. As expected, the modification severely inhibited tRNA selection by the ribosome as reflected by the three orders of magnitude drop in the observed rate of peptide bond formation and the tenfold decrease in the endpoint of the reaction (*Figure 2C*). This is consistent with a previous study showing that m$^1$A in mRNA significantly decreased protein-synthesis yield in S30 extracts (*You et al., 2017*). Interestingly, while the observation that m$^1$A severely inhibits decoding is expected, our finding that the addition of paromomycin does not suppress this effect at all (*Figure 3*) is unanticipated. Since paromomycin allows the incorporation of near-cognate aa-tRNAs (*Pape et al., 2000*; i.e. those that have one mismatch between their anticodon and the A-site codon), this observation suggests that the modification is so disruptive that it affects base-pairing between the neighboring nucleotides. Consequently, all aa-tRNAs are recognized as non cognates, or as having more than one mismatch (*Figure 4*).

Having confirmed that m$^1$A-modified mRNAs are highly inhibitory to tRNA selection in vitro, we then sought to investigate how bacterial cells deal with alkylated RNA, which includes m$^1$A-modified ones, in vivo. We hypothesized that *trans* translation is likely to be activated in response to stalls caused by alkylation damage to mRNA. To add support for this hypothesis, we used *E. coli* strains harboring a modified *ssrA* gene (encoding tmRNA), for which the encoded ssrA-degradation tag is altered to one that codes for a His tag (*Moore and Sauer, 2005*). In these strains, the addition of His tags, as evaluated by western blot analysis, can be used to assess tmRNA activation. In complete agreement with our model, we observe a significant increase in His-tagging and hence increased *trans*-translation activity upon cellular treatment with alkylating agents (*Figure 5A*). We also showed that this activity is independent of active transcription (*Figure 5*) and hence is likely to be initiated as a result of direct damage to mature mRNAs. Future experiments aimed at characterizing the identity of the tagged peptides, using mass-spectrometry approaches, for example, are likely to reveal important insights into the specificity of tmRNA tagging as well as that of the antibody recognition of tagged products. Since MMS is known to modify A and C, at least in a way that affects their Watson-Crick-base pairing properties, we expect tagging to occur preferentially on codons enriched for these two nucleotides.

Notably, because bacterial mRNAs do not contain poly-A tails like their eukaryotic counterparts, it is difficult to purify mRNA away from rRNA and tRNA; therefore, we cannot be certain that the observed effects on translation are exclusively due to mRNA damage. However, there exist several compelling reasons to support the hypothesis that the tmRNA response is primarily due to mRNA damage. For one, *trans* translation has been studied almost exclusively in the context of defective mRNAs. It is difficult to imagine how a defective ribosome would still be capable of performing peptidyl transfer only with tmRNA. Second, the folding of rRNA as well as its association with ribosomal proteins is thought to make it a poor target for alkylative damage (*Simms and Zaher, 2016*). Specifically, the rRNA residues responsible for monitoring the base pairing in the decoding center are not exposed; therefore, it is unlikely that they are damaged by the agents at a high incidence (*Clemons et al., 1999*). Additionally, tRNAs are susceptible to alkylative damage, but only 3 out of an average of 76 nucleotides directly participate in base pairing with the codon. This reduces the probability that damaged nucleotides in tRNA cause the observed ribosomal stalling. Furthermore, unless the P-site tRNA is damaged, tmRNA is unlikely to be able to sense defects to the tRNA pool. Finally, the CCA-adding enzyme in *E. coli* has been shown to discriminate against tRNA backbone damage (*Hou, 2010*); without CCA ends these tRNAs are not aminoacylated and cannot participate in peptidyl transfer.

Several studies have shown that bacteria lacking a functional *trans*-translation pathway do not recover as efficiently after cellular stress, including metabolic and oxidative stress (*Nishikawa et al., 2006*; *Karzai et al., 1999*). We observe that alkylative stress also causes a delayed recovery period in cells lacking tmRNA, likely because they are unable to efficiently rescue stalled ribosomes and resume growth (*Figure 7*). The ΔssrA cells are likely able to eventually resume growth after alkylative damage because of the existence of several alternative ribosome-rescue factors. One factor, known as alternative ribosome-rescue factor A (ArfA) works by recruiting RF2 to hydrolyze the peptidyl-tRNA and release the ribosome (*Shimizu, 2012*; *Chadani et al., 2012*). This factor acts as a backup for *trans*-translation, as its expression increases when tmRNA activity is limited (*Garza-Sánchez et al., 2011*; *Chadani et al., 2011a*). Another factor that can release stalled ribosomes is ArfB, although it does not appear to function solely as a backup for tmRNA and its physiological function remains to be elucidated (*Chadani et al., 2011b*; *Handa et al., 2011*). Regardless, these

alternative ribosome rescue factors in *E. coli* are likely responsible for the eventual recovery we observe, and future studies should be aimed at exploring their role in the alkylation-stress response.

# Materials and methods

## Key resources table

| Reagent type (species) or resource | Designation | Source or reference | Identifiers | Additional information |
|---|---|---|---|---|
| Gene (*Escherichia coli*) | ssrA | UniProt | P0A832 | |
| Gene (*Escherichia coli*) | smpB | UniProt | P0A832 | |
| Gene (*Escherichia coli*) | ClpX | UniProt | P0A6H1 | |
| Gene (*Escherichia coli*) | ClpP | UniProt | P0A6G7 | |
| Gene (*Escherichia coli*) | Lon | UniProt | P0A9M0 | |
| Strain, strain background (*Escherichia coli*) | MG1655 | PMID:6271456 | | Wild type |
| Strain, strain background (*Escherichia coli*) | X90 | PMID:2651442 | | *ara∆(lac-pro) nalA argE*(Am) *rif thi-1/F' lacl^q lac^+ pro^+* |
| Strain, strain background (*Escherichia coli*) | SM20 | PMID:16194232 | | Gift from Sean Moore; X90, *∆ssrA, cam* |
| Strain, strain background (*Escherichia coli*) | SM694 | PMID:16194232 | | Gift from Sean Moore; X90, *ssrA::his₆ - kan* |
| Strain, strain background (*Escherichia coli*) | SM876 | PMID:16194232 | | Gift from Sean Moore; X90, *ssrA::his₆ - kan, clpPX-lon::cam* |
| Strain, strain background (*Escherichia coli*) | SKEC4 | Other | | Gift from Kazuki Saito and Allen Buskirk; MG1655, *∆ssrA, ∆smpB, kan* |
| Strain, strain background (*Escherichia coli*) | ET1 | This Paper | | MG1655, *ssRA::his₆ - kan* |
| Antibody | Goat polyclonal anti-mouse IgG-HRP | Thermo Scientific | Cat#: 31430; RRID:AB_228307 | WB: (1:10000) |
| Antibody | Goat polyclonal anti-mouse IgG-HRP | Thermo Scientific | Cat#: 31460; RRID:AB_228341 | WB: (1:10000) |
| Antibody | Rabbit polyclonal anti-RF2-His | PMID:22000017 | | WB: (1:1000) |
| Antibody | Mouse monoclonal anti-6× His tag antibody [HIS.H8] | Abcam | Cat#: ab18184; RRID:AB_444306 | WB: (1:2500) |
| Antibody | Mouse monoclonal anti-His-tag antibody (H-3) | Santa Cruz Biotechnology | Cat#: sc-8036; RRID:AB_627727 | WB: (1:2500), HRP-conjugated |
| Antibody | Mouse monoclonal THE His Tag Antibody | Genscript | Cat#: A00186; RRID:AB_914704 | WB: (1:2500) |
| Antibody | Mouse monoclonal anti-Ad (ADA-1) | Santa Cruz Biotechnology | Cat#: sc-53152; RRID:AB_772487 | WB: (1:500) |

*Continued on next page*

*Continued*

| Reagent type (species) or resource | Designation | Source or reference | Identifiers | Additional information |
|---|---|---|---|---|
| Antibody | Rabbit polyclonal anti-RecA | Abcam | Cat#: ab63797; RRID:AB_1142554 | WB: (1:10000) |
| Antibody | Rabbit polyclonal anti-m6A | Synaptic Systems | Cat#: 202 003; RRID:AB_2279214 | IB: (1:5000) |
| Commercial assay or kit | SuperSignal West Pico chemiluminescent substrate | Thermo Scientific | Cat#: 30480 | |
| Chemical compound, drug | Methyl methanesulfonate | Millipore Sigma | Cat#: 129925 | |
| Chemical compound, drug | 1-methyl-3-nitro-1-nitrosoguanidine | Fisher | Cat#: M05275G | Mfr: TCI America |
| Chemical compound, drug | Ciprofloxacin | Millipore Sigma | Cat#: 17850 | |
| Chemical compound, drug | Mitomycin C | Millipore Sigma | Cat#: M4287 | |
| Chemical compound, drug | Rifampicin | Millipore Sigma | Cat#: R3501 | |
| Chemical compound, drug | Nuclease $P_1$ from *Penicillium citrinum* | Sigma | Cat#: N8630 | |
| Chemical compound, drug | Alkaline phosphatase, calf intestinal (CIP) | NEB | Cat#: M0290 | |
| Chemical compound, drug | Paromomycin sulfate | VWR | Cat#: AAJ61274-06 | |
| Chemical compound, drug | EasyTag L-[$^{35}$S]-Methionine | PerkinElmer | Cat#: NEG709A005MC | |
| Chemical compound, drug | Transfer ribonucleic acid: glutamic acid specific | Chemical Block | | |
| Chemical compound, drug | Transfer ribonucleic acid: N-Formyl-methionine specific | Chemical Block | | |
| Chemical compound, drug | tRNA from *E. coli* MRE 600 | Millipore Sigma | Cat#: 10109541001 | Mfr: Roche |
| Chemical compound, drug | $N^6$-methyladenosine | Berry and Associates | Cat#: PR3732 | |
| Chemical compound, drug | $O^6$-methylguanosine | Berry and Associates | Cat#: PR3757 | |
| Chemical compound, drug | 1-methyladenosine | Cayman chemical | Cat#: 16937 | |
| Chemical compound, drug | N1-methylguanosine | Carbosynth | Cat#: NM08574 | |
| Chemical compound, drug | N3-methylcytidine | Carbosynth | Cat#: NM05757 | |
| Chemical compound, drug | N7-methylguanosine | Carbosynth | Cat#: NM08037 | |
| Chemical compound, drug | Adenosine | Fisher | Cat#: AC164040050 | Mfr: ACROS Organics |
| Chemical compound, drug | Guanosine | Fisher | Cat#: AC411130250 | Mfr: ACROS Organics |
| Chemical compound, drug | Cytidine | Fisher | Cat#: AC111810100 | Mfr: ACROS Organics |
| Chemical compound, drug | Uridine | Tokyo Chemical Industry | Cat#: U0020 | |

*Continued on next page*

*Continued*

| Reagent type (species) or resource | Designation | Source or reference | Identifiers | Additional information |
|---|---|---|---|---|
| Sequence-based reagent | $m_1A$ oligo | Midland Certified Reagent Company | | C AGA GGA GGU AAA AAA AUG G(1-methyl-A) A UUG UAC AAA |
| Sequence-based reagent | Control oligo | This Paper | | C AGA GGA GGU AAA AAA AUG GAA UUG UAC AAA |
| Software, algorithm | Graphpad Prism | GraphPad Prism (http://www.graphpad.com/) | RRID:SCR_002798 | |
| Software, algorithm | Agilent MassHunter WorkStation - Qualitative Analysis for GC/MS | Agilent (https://www.agilent.com/en/products/software-informatics/masshunter-suite/masshunter-qualitative-analysis-gcms) | RRID:SCR_016657 | Version B.08.00 |
| Software, algorithm | Image Quant TL | Cytivia (https://www.cytivalifesciences.com/en/is/shop/protein-analysis/molecular-imaging-for-proteins/imaging-software/imagequant-tl-8-2-image-analysis-software-p-09518) | RRID:SCR_018374 | Version 8.2 |
| Software, algorithm | Fiji | Fiji (http://fiji.sc) | RRID:SCR_002285 | |
| Software, algorithm | Quantity One 1-D Analysis Software | Bio-Rad (http://www.bio-rad.com/en-us/product/quantity-one-1-d-analysis-software) | RRID:SCR_014280 | |

## Strains

Strains were either derivatives of *E. coli* MG1655 (F- lambda- ilvG- rfb-50 rph-1) or X90 (araΔ(*lac-pro*) nalA argE(Am) *rif thi-1*/F' lacI$^q$ lac$^+$ pro$^+$; *Parsell and Sauer, 1989*). The following strains SM694 (X90, *ssrA::his$_6$ - kan*), SM876 (X90, *ssrA::his$_6$ - kan, clpPX-lon::cam*), and SM20 (X90, Δ*ssrA, cam*) were a gift from Dr. Sean Moore (*Moore and Sauer, 2005*). The SKEC4 strain (MG1655, Δ*ssrA*, Δ*smpB, kan*) was a gift from Dr. Allen Buskirk. P1 transduction (*Thomason et al., 2007*) was used to introduce kan$^R$-linked tmRNA-H$_6$ into MG1655.

## Western blot analysis

To prepare total protein for Western blot analysis, *E. coli* were collected, washed with LB, and resuspended in 2 × SDS loading dye. The resuspension volume was adjusted to normalize for OD$_{600nm}$ of the culture at the time of collection. Total protein was separated by SDS-PAGE and transferred to PVDF membrane in 1 × transfer buffer (25 mM Tris, 192 mM glycine, 20% methanol) in a wet apparatus. After transfer, the membrane was blocked for 1 hr in PBST (3.2 mM Na$_2$HPO$_4$, 0.5 mM KH$_2$PO$_4$, 1.3 mM KCl, 135 mM NaCl, 0.05% Tween 20, pH 7.4) containing 5% w/v powdered milk. The membrane was then washed with PBST and incubated with the primary antibody overnight at 4° C. The following dilutions of primary antibodies were used: 1:2500 anti-His (Abcam unless otherwise specified), 1:500 anti-Ada (Santa Cruz Biotechnologies), 1:10,000 anti-RecA (Abcam), and 1:1000 anti-RF2 (purified as described in *Zaher and Green, 2011*). The blot was then washed three times for 5 min and then incubated with the corresponding HRP-conjugated secondary antibody (1:10,000) (ThermoFisher) in PBST for 1 hr . After washing three times for 5 min, the membrane was treated with an HRP-reactive chemiluminescent reagent (Pierce ECL Western Blotting Substrate). Quantity One software was utilized to quantify Western blots.

## Treatment of *E. coli* with damaging agents

For all Western blot analyses, *E. coli* were treated with the following concentrations of damaging agents: 0.1% MMS (Sigma-Aldrich), 5 μg/mL MNNG (Tokyo Chemical Industry), 50 μg/mL ciprofloxacin (Sigma-Aldrich), or 6 μg/mL mitomycin C (Sigma-Aldrich). To determine the treatment time that generated significant tmRNA activity and Ada activation, MG1655 cells containing tmRNA-His$_6$ were

grown from OD 0.05 to mid-log phase (OD 0.3–0.4) and treated with MMS for several time points. The resulting total protein was analyzed via Western blot (*Figure 5—figure supplement 5*). We observed significant tmRNA activity and Ada activation after a 20 min treatment, which is the treatment time we utilized for the remaining samples analyzed via Western blot.

To determine the optimal length of time for rifampicin pre-treatments, we treated MG1655 cells containing tmRNA-His$_6$ with 6 ug/mL rifampicin for several time points followed by 20 min treatments with either MMS or ciprofloxacin. The resulting protein was analyzed via Western blot (*Figure 5—figure supplement 6*). We observed significant decreases in Ada activation in the MMS-treated samples and significant decreases in tmRNA activity and RecA activation in the ciprofloxacin-treated samples after 10 s of rifampicin pre-treatment. We utilized a 10 to 45 s rifampicin pre-treatment time for the remaining samples analyzed via Western blot.

## Quantification of nucleosides via liquid chromatography – mass spectrometry

Overnight cultures of MG1655 *E. coli* were diluted to OD$_{600nm}$ 0.05 in LB and grown to an OD$_{600nm}$ of 0.3–0.4 at 37°C before 20 min treatment with either 0.1% MMS or 5 µg/mL MNNG. RNA was isolated using a hot phenol method as previously described (*Simms et al., 2017b*). About 10 µg of total RNA was digested by P1 nuclease (Sigma-Aldrich, 10 Units) at 50°C overnight. The pH was adjusted by adding Tris pH 7.5 to a final concentration of 100 mM before calf intestinal phosphatase (NEB) was added to a final concentration of 0.2 U/µL, and the reaction was further incubated for 1 hr at 37°C to convert 5′-monophosphates to nucleosides. The samples were diluted to 150 µL and filtered (0.22 µm pore size) before injecting 10 µL into an Agilent 1290 Infinity II UHPLC connected to an Agilent 6470 Triple Quadrupole mass spectrometer. Nucleosides were separated on a Zorbax Eclipse Plus C18 column (2.1 × 50 mm × 1.8 micron) and then analyzed using multiple-reaction monitoring in positive-ion mode. Calibration curves were generated with known concentrations of standards. Unmodified nucleosides were monitored by absorbance at 260 nm. Modified nucleosides were monitored by MRM. The retention times and mass transitions of each nucleoside are listed in *Supplementary file 1*. Free unmodified A, G, and C standards were purchased from Acros Organics and U was purchased from Tokyo Chemical Industry. Free modified nucleosides m$^7$G, m$^1$G, and m$^3$C were purchased from Carbosynth, m$^6$G, and m$^6$A were purchased from Berry's Associates, and m$^1$A was purchased from Cayman Chemical Company. Data was analyzed using Agilent qualitative analysis, Excel, and Graphpad Prism software.

## Charging of aminoacyl-tRNA

[$^{35}$S]-fMet-tRNA$^{fMet}$ was prepared as previously described (*Walker and Fredrick, 2008*). The tRNAs were aminoacylated by incubating total tRNA mix (Roche) at 150 µM with the appropriate amino acid (0.4 mM), tRNA synthetase (~5 µM), and ATP (2 mM) in a charging buffer composed of 100 mM K-HEPES (pH 7.6), 20 mM MgCl$_2$, 10 mM KCl, and 1 mM DTT. After a 30 min incubation at 37°C, the aa-tRNAs were purified by phenol/chloroform extraction, ethanol precipitated, and resuspended in 20 mM KOAc (pH 5.2) and 1 mM DTT.

## Formation of ribosome initiation complexes

Protocols were performed as previously described (*Pierson et al., 2016*). All initiation complex (IC) formation and peptidyl transfer reactions were performed in 1 × polymix buffer (*Jelenc and Kurland, 1979*; *Gromadski and Rodnina, 2004a*), composed of [95 mM KCl, 5 mM NH$_4$Cl, 5 mM Mg(OAc)$_2$, 0.5 mM CaCl$_2$, 8 mM putrescine, 1 mM spermidine, 10 mM K$_2$HPO$_4$ (pH 7.5), 1 mM DTT]. In order to generate ICs, 70S ribosomes (2 µM), IF1, IF2, IF3, [$^{35}$S]-fMet-tRNA$^{fMet}$ (3 µM each), mRNA (6 µM), and GTP (2 mM) were incubated in 1 × polymix buffer at 37°C for 30 min. The initiation complexes were purified from free tRNAs and initiation factors over a 500 µL sucrose cushion composed of 1.1 M sucrose, 20 mM Tris-HCl pH 7.5, 500 mM NH$_4$Cl, 0.5 mM EDTA, and 10 mM MgCl$_2$. The mixture was spun for 2 hr at 287,000 × g at 4°C, and the pellet was resuspended in 1 × polymix buffer and stored at −80°C. The fractional radioactivity that pelleted was used to determine the concentration of IC.

Modified mRNAs containing m$^1$A used in the IC formation reaction were purchased from The Midland Certified Reagent Company, and its sequence is as follows: C AGA GGA GGU AAA AAA

AUG G(1-methyl-A)A UUG UAC AAA. The unmodified control mRNA was transcribed from a dsDNA template using T7 polymerase and purified via denaturing PAGE (*Zaher and Unrau, 2004*).

## Kinetics of peptidyl transfer

In order to exchange bound GDP for GTP, EF-Tu (30 µM final) was initially incubated with GTP (2 mM final) in 1 × polymix buffer for 15 min at 37°C. The mixture was then incubated with aminoacyl-tRNAs (~6 µM) for 15 min at 37°C to form ternary complexes (TC). For reactions performed in the presence of paromomycin, 10 µg/mL final of the antibiotic were added to the mixtures. Kinetics assays were also performed using *trans*-translation quaternary complexes (QCs), which were formed by incubating Ala-tmRNA$^{Ala}$ with SmpB, EF-Tu, and GTP in 1 × polymix for 15 min at 37°C. The TC or QC mixture was then incubated with an equivalent volume of IC at 37°C either using an RQF-3 quench-flow instrument or by hand. KOH to a final concentration of 500 mM was used to stop reactions at different time points. Dipeptide products and free fMet were separated using cellulose TLC plates that were electrophoresed in pyridine-acetate at pH 2.8 (50). TLC plates were then exposed to a phosphor screen overnight, after which they were imaged using a Personal Molecular Imager (PMI) system. The images were quantified, and the fraction of dipeptide fMet at each time point was used to determine the rate of peptide bond formation using GraphPad Prism software.

## S30 in vitro translation

Protocol for preparing S30 extract was adapted from *Kigawa et al., 2004*. 2 L of SM876 were grown in YT medium (1.6% tryptone, 1% yeast extract, 0.5% NaCl) at 37°C from an overnight culture. Cells were harvested at $OD_{600nm}$ ~2 and washed in buffer 1 (10 mM Tris/OAc pH 8.2, 60 mM KOAc, 14 mM Mg(OAc)$_2$, 1 mM DTT, 7 mM β-ME) and then buffer 2 (buffer 1 minus β-ME). The resulting pellet was weighed and resuspended in 1.3 mL buffer per 1 g of wet cell pellet. Cells were lysed by passing them through a French press three times. The resulting lysate was clarified twice by centrifugation at 30,000 × g for 30 min at 4°C. The supernatant was then transferred to a 15 mL conical tube and incubated with 0.15 × volume of preincubation buffer (300 mM Tris/OAc pH 7.6, 10 mM Mg (OAc)$_2$, 10 mM ATP, 80 mM phosphoenolpyruvate, 5 mM DTT, 40 µM of each amino acid, 8 U/mL pyruvate kinase) for 90 min at 37°C in the dark. The lysate was dialyzed (mwco: 3500 Da) overnight in buffer 2 at 4°C and dialyzed again the next day for 1 hr in fresh buffer. Dialyzed lysate was centrifuged at 4000 × g for 10 min at 4°C and the supernatant aliquoted and flash frozen in liquid nitrogen.

mRNAs templates for in vitro were generated by T7 RNA polymerase as previously described (*Zaher and Unrau, 2004*). mRNAs were subjected to two rounds of phenol-chloroform extractions followed by ethanol precipitation. To complete the purification process, the transcripts were applied to P-30 gel filtration spin columns (Bio-Rad) to remove abortive short transcripts. MMS treatment of mRNA was conducted by incubating ~35 µg of RNA with 0.5% MMS in a total volume of 100 µL for 15 min followed by phenol/chloroform extraction and ethanol precipitation.

A typical 10 µL in vitro translation reaction contained the following: 3 µg of RNA, 1.2 µL Buffer 1, 2.4 µL S30 extract, 4 µL of S30 Premix Plus from the Promega S30 T7 High-Yield Protein Expression System kit, and 2 µCi L-[$^{35}$S]-Methionine. Reactions were incubated at 37°C for 1 hr and samples were resolved using 15% bis-tris gels followed by transfer to PVDF membranes. The analysis was done using autoradiography on a Typhoon Phosphorimager and Western blotting.

## Analysis of mRNA modification by immunoblotting

About 50 mL of MG1655 and Δ*ssrA* MG1655 cells were grown to $OD_{600nm}$ of 0.4. At this point a 10 mL aliquot was harvested, quickly centrifuged and the cell pellet flash frozen. To the remaining culture, 40 µL of MMS was added (final concentration of 0.1%) and incubated for 15 min. A 10 mL aliquot was collected as before. The rest of the cells were centrifuged, washed with fresh LB, and allowed to recover for 10 min, at which point a third aliquot was removed. Total RNA was isolated from samples using the hot phenol method as described above. Samples were resolved using denaturing agarose gel electrophoresis. At this stage, gels were either promptly transferred to a nitrocellulose membrane in 10 × SSC buffer or incubated in buffer containing 50 mM NaOH, 1.5 M NaCl to convert m$^1$A to m$^6$A before transfer. Following the transfer, samples were cross linked by

UV treatment, blocked with milk in TBST (20 mM Tris-HCL pH 7.5, 150 mM NaCl, 0.1% Tween 20) before incubation with anti $m^6A$ antibody (1:5,000; Synaptic Systems).

## Alkylative damage recovery assays

Overnight cultures of MG1655 and $\Delta ssrA$ MG1655 cells were diluted to $OD_{600nm}$ 0.05 and grown to 0.3–0.4 at 37˚C before treating with either 0.5% MMS or 20 µg/mL MNNG for 20 min. The $OD_{600nm}$ of the cells was recorded at the time of collection, and the samples were washed twice with LB and resuspended in an adjusted volume of LB. Cells were diluted to an $OD_{600nm}$ of 0.005 at 100 µL final volume in a 96-well plate. Plates were shaken at 37˚C for 20 hr in a BioTek Eon microtiter plate reader which measured the $OD_{600nm}$ of each well every 10 min.

## Spot assays for viability analysis

X90 and SM20 *E. coli* were grown from $OD_{600nm}$ 0.05 to OD 0.3 at 37˚C before treating with either 0.1% or 0.5% MMS. At each time point, an aliquot of the culture was removed, washed with LB, and then serially diluted 1:10 eight times. About 4 µL of each dilution was spotted on an LB plate. The plates were imaged, and colonies were counted using the Colony Counter plugin on ImageJ.

## Acknowledgements

We thank Sean Moore for his gift of the tmRNA-His$_6$ strains, and Allen Buskirk for the tmRNA-deletion ones. We also thank Carrie Simms and members of the Zaher laboratory for useful discussions on earlier versions of this manuscript. This work was supported by a grant from the National Institutes of Health to HSZ (R01GM112641).

## Additional information

### Funding

| Funder | Grant reference number | Author |
| --- | --- | --- |
| National Institutes of Health | R01GM112641 | Hani S Zaher |

The funders had no role in study design, data collection and interpretation, or the decision to submit the work for publication.

### Author contributions

Erica N Thomas, Conceptualization, Data curation, Formal analysis, Investigation, Methodology, Writing - original draft, Writing - review and editing; Kyusik Q Kim, Data curation, Formal analysis, Investigation, Methodology, Writing - review and editing; Emily P McHugh, Thomas Marcinkiewicz, Data curation, Formal analysis; Hani S Zaher, Conceptualization, Resources, Data curation, Formal analysis, Supervision, Funding acquisition, Investigation, Methodology, Writing - original draft, Project administration, Writing - review and editing

### Author ORCIDs

Kyusik Q Kim (iD) https://orcid.org/0000-0002-1832-2070
Hani S Zaher (iD) https://orcid.org/0000-0002-7424-3617

### Decision letter and Author response

Decision letter https://doi.org/10.7554/eLife.61984.sa1
Author response https://doi.org/10.7554/eLife.61984.sa2

## Additional files

### Supplementary files

• Supplementary file 1. Mass transitions, retention times, and collision energies for nucleoside standards.

• Transparent reporting form

### Data availability

All data generated or analysed during this study are included in the manuscript and supporting files.

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
