## [Decision Letter]

**Acceptance summary:**

This paper reveals that alkylation of mRNA, which impairs the speed and accuracy of protein synthesis, activates the bacterial transfer-messenger RNA (tmRNA) ribosome rescue pathway. This is an exciting finding because traditionally tmRNA has been known to work on truncated transcripts (lacking a stop codon); this work thus reveals a new expanded role for ribosome rescue on chemically damaged mRNAs.

**Decision letter after peer review:**

[Editors’ note: the authors submitted for reconsideration following the decision after peer review. What follows is the decision letter after the first round of review.]

Thank you for submitting your work entitled "Alkylative damage of mRNA leads to ribosome stalling and rescue by trans translation in bacteria" for consideration by *eLife*. Your article has been reviewed by three peer reviewers, one of whom is a member of our Board of Reviewing Editors, and the evaluation has been overseen by a Senior Editor. The reviewers have opted to remain anonymous.

Our decision has been reached after consultation between the reviewers. Based on these discussions and the individual reviews below, we regret to inform you that your work will not be considered further for publication in *eLife* at this point.

We expect that the experiments needed to address the reviewer concerns, see below, will take longer than the typical two-month period provided by the journal. That being said, we hope that you will consider the comments below and in the full reviews, and that you will consider re-submitting this work to *eLife* if you can address the key concerns.

In their consultation and in the reviews appended below, all three reviewers commented positively on the novelty and interest of the work, and on the use of both in vivo and in vitro experiments to assess the impact of RNA alkylation. The kinetic analyses show convincingly that alkylation of the second base of a codon significantly impaired decoding, resulting in reduced rates and yields of protein synthesis. The most intriguing finding is the potential role of the ribosome rescue factor tmRNA, which typically works on truncated mRNAs, in rescuing ribosomes from the damaged mRNAs.

Despite their enthusiasm toward these findings, the reviewers expressed several concerns; most notably in trying to link the in vitro and in vivo data. They felt that stronger evidence is needed linking tmRNA to rescue of ribosomes on alkylated mRNAs in cells. Additional evidence is needed to rule out the possibility that the tmRNA is targeting truncated mRNAs transcribed from damaged DNA rather than targeting alkylated mRNAs in the cell. For example, by assessing the role of tmRNA on electroporated unmodified or alkylated mRNAs. Relatedly, examination of turnover of the alkylated mRNAs is needed to assess the relative importance of tmRNA versus other ribosome rescue factors.

Reviewer #1:

This paper from the Zaher lab describes studies examining the impact on translation caused by alkylation of mRNA caused by MMS or MNNG treatments. in vitro, m1A modification of the middle base of an A site codon, substantially impaired the rate and extent of peptide formation. Based on this defect in peptide bond formation, the researchers examined whether the stalled translation product was a substrate for rescue by the tmRNA system. First, they found that the m1A modification did not substantially impair trans-translation in vitro. Next, using a tagged tmRNA, they found that alkylative damage enhanced tagging of many cellular proteins in vivo, suggesting that alkylative damage generates tmRNA substrates in the cell. Finally, the authors found that disabling the trans-translation rescue pathway sensitized *E. coli* cells to alkylation of mRNAs.

Overall, this is an interesting story and complements previous work showing that alkylative mRNA damage in eukaryotes triggers ribosome quality control (RQC) pathways including degradation of the stalled nascent peptide. The work in this paper indicates that a parallel quality control pathway is functional in bacteria to eliminate the stalled translation products caused by mRNA alkylation.

Specific comments:

1) In Figure 1A, why do the levels m6A decrease in the presence of MMS and MNNG?

2) In the last sentence of Results paragraph two, it is stated that the base level of m1G is at least 200-fold higher than that of m1A. However, this is not obvious based on Figures 1A and 1B and the different Y-axes used; in fact, it looks like m1A levels are 4-5-fold higher than m1G.

3) Results, third paragraph: the citations to different panels in Figure 1 are incorrect. Second line should refer to Figure 1D and 1E; third line to Figure 1F and panel C (m7G) is never cited in the text.

4) Figure 3: there is no positive control to show that the antibiotic was functional in these experiments

5) Figure 4 is described as testing the 20 possible ternary complexes. However, the number of possible ternary complexes is not dictated by the amino acid, but by the tRNAs (around 46 different tRNAs). Since the experiment was examining the ability of different tRNAs to decode the modified codon, the number of possible tRNAs (with distinct anticodons) is more relevant than the different amino acids.

6) Are the modified (alkylated) mRNAs turned over? Do the levels of modified nucleotides increase in the ∆-ssrA cells treated with MMS?

Reviewer #2:

This manuscript by Zaher and co-workers describes the effect of alkylation agents on protein synthesis in *E. coli*. First they characterize which bases in total RNA are modified by the alkylating agents MMS and MNNG. An earlier report using in vitro extracts (You, Dai and Wang, 2017) showed that m1A and other modifications in mRNA severely reduce protein synthesis in *E. coli*. Here the authors use purified components and pre-steady state kinetic methods to demonstrate that the rate and overall yield of peptide bond formation is dramatically reduced when m1A is found at the second position of the A-site codon. There is a severe decoding defect as evidenced by the fact that addition of antibiotics known to induce miscoding with near-cognate (but not non-cognate) tRNAs do not improve decoding of a codon with an alkylated base. These biochemical data are compelling although they provide only a small step forward in terms of understanding how ribosomes deal with alkylated messages.

The authors then ask how *E. coli* cells deal with ribosomes stalled on alkylated bases. They show that the activity of tmRNA, the main ribosome rescue factor, is increased when cells are treated with alkylating agents. This is an exciting finding because traditionally tmRNA has been known to work on truncated transcripts (lacking a stop codon); a role for ribosome rescue on chemically damaged mRNAs is new. My enthusiasm is tempered by three concerns: 1) they have not completely ruled out that alkylating agents damaging DNA might create truncated transcripts (the usual substrate for tmRNA) that could explain the increase in tmRNA activity without tying it to alkylation of mRNA. 2) There is a major gap in evidence and understanding between the in vitro biochemistry and the proteome-wide tmRNA activity detected in vivo. 3) The physiological importance of tmRNA in responding to alkylating conditions seems to be quite minor: yes, there is a one hour lag in recovery when tmRNA is lost, but wild-type and tmRNA minus cells survive this treatment equally well.

1) A major concern is transcription from damaged DNA could make truncated messages that increase tmRNA tagging, meaning that the signal may not come from ribosome stalling on damaged mRNAs. The authors try to use rifampicin to block transcription, but it doesn't fully work. The ciprofloxacin His6 signal is not eliminated by Rif pre-treatment and Ada is still being made: this means that transcription still is happening.

An alternative strategy would be to work directly at the RNA level: electroporate mRNA encoding a reporter with a N-terminal FLAG tag. Pull down all the reporter protein of any length with anti-FLAG antibodies and see how much of it is tagged with His6 by the altered tmRNA. The mRNA could be treated with various amounts of alkylation prior to electroporation. This experiment would rule out any effects of DNA damage. It could be done in S30 extracts (where tmRNA is active) if electroporation proves to be impossible.

2) One of the nice features of the Zaher lab's earlier paper in 2018 (Nat. Comm.) about nucleotide damage and ribosome rescue in yeast was the link to RNA decay. In bacteria, tmRNA-mediated ribosome rescue leads to mRNA degradation by RNase R. In contrast, in the absence of tmRNA, rescue by ArfA does not promote mRNA degradation. To more conclusively show that tmRNA is acting on messages with alkylated damage, the authors could perform a pulse chase experiment, first labeling mRNA (the pulse) and then during the chase period, adding a short alkylation treatment, then observing the rates of turnover of labeled mRNA. They could compare the rates in wild-type cells vs those lacking tmRNA or RNase R. This experiment would rule out transcription problems (since the RNA is synthesized before alkylating agents are added). It might also show that alkylation affects mRNA decay and tie tmRNA activity to rescue on these transcripts by its effects on their decay.

Reviewer #3:

This study from the Zaher lab addresses the possible effects of alkylating agents on translation, in particular as a result of m1A adducts generated in mRNA. The in vitro and in vivo assays show significant effects on translation, and the authors also investigated how tmRNA might be part of the cellular response to alkylation. The study is fairly interesting, if not entirely unsurprising in its findings. The role of tmRNA is based on prediction rather than a discovery-based approach, resulting in the study providing an ultimately limited insight into the cellular response to alkylation at the level of translation.

Numbered summary of any substantive concerns

1) Why were "sub-optimal substrates" used for the in vitro trans translation assays?

2) The possible roles of ArfA and ArfB were not experimentally investigated, it is possible they also protect against m1A adducts (as is mentioned in the Discussion) or is this a tmRNA-specific effect? Some simple growth assays in the appropriate genetic backgrounds might be informative.

3) As the authors concede for the in vivo experiments, possible effects of damage to the ribosomes cannot be excluded. The argument in the discussion is not convincing as to alkylation could not be assessed, if only globally initially, for 30S and 50S subunits.

[Editors’ note: further revisions were suggested prior to acceptance, as described below.]

Thank you for submitting your article "Alkylative damage of mRNA leads to ribosome stalling and rescue by trans translation in bacteria" for consideration by *eLife*. Your article has been reviewed by three peer reviewers, one of whom is a member of our Board of Reviewing Editors, and the evaluation has been overseen by Gisela Storz as the Senior Editor. The reviewers have opted to remain anonymous.

The reviewers have discussed the reviews with one another and the Reviewing Editor has drafted this decision to help you prepare a revised submission.

Summary:

This work from the Zaher lab describes studies examining the impact on translation caused by alkylation of mRNA caused by MMS or MNNG treatments in E coli. in vitro, m1A modification of the middle base of an A site codon, substantially impaired the rate and extent of peptide formation. There is a severe decoding defect as evidenced by the fact that addition of antibiotics known to induce miscoding with near-cognate (but not non-cognate) tRNAs do not improve decoding of a codon with an alkylated base. Based on this defect in peptide bond formation, the researchers examined whether the stalled translation product was a substrate for rescue by the tmRNA system. First, they found that the m1A modification did not substantially impair trans-translation in vitro. Next, using a tagged tmRNA, they found that alkylative damage enhanced tagging of many cellular proteins in vivo, suggesting that alkylative damage generates tmRNA substrates in the cell. Finally, the authors found that disabling the trans-translation rescue pathway sensitized *E. coli* cells to alkylation of mRNAs. This is an exciting finding because traditionally tmRNA has been known to work on truncated transcripts (lacking a stop codon); a role for ribosome rescue on chemically damaged mRNAs is new and opens questions of other modifications and their potential in quality control.

In this re-submission, the authors address concerns raised regarding their original submission. Of note, they present in vitro data in new Figure 7 showing that MMS-treated mRNA is not as well translated as the untreated mRNA. Moreover, the MMS-treated mRNA produces more tma-tagged products indicating that the damaged mRNA is in fact a tma substrate. In addition, in new Figure 6—figure supplement 1 they provide evidence of increased levels of modified mRNAs in cells lacking ssrA. This latter result is consistent with their hypothesis that the damaged mRNAs are turned over following tma tagging of the translation product.

Revisions:

1) It is striking that the pattern of tma-tagged products is not altered upon MMA treatment. This is noted in vivo in Figure 6 where the level abundance of tagged products are increased in MMS-treated versus untreated cells, but the pattern of products on the gel are not changed. The new data in Figure 7 shows the same result in vitro. The untreated and treated mRNAs yield the same pattern of tma-tagged products. The authors propose that this is due to the specificity of the antibody and they show that different antibodies reveal different patterns. While the authors' hypothesis seems reasonable at this time, and no additional experiments are requested, we feel that this unexpected result is commented on in this review.

2) Is it appropriate to refer to Figure 2 as testing "all 20 possible aa-tRNA isoacceptors"? While perhaps formally correct, they did test one isoacceptor for each amino acid, the relevant tRNA attribute in the assay is the anticodon. I would be more comfortable if they stated that they tested 20 different tRNAs with unique anticodons. Later in the paragraph, they state that "no significant dipeptide accumulation occurred in the presence of any aa-tRNA". I think this should read "no significant dipeptide accumulation occurred in the presence of any of the tested aa-tRNAs"

3) One concern is that transcription from damaged DNA could make truncated messages that increase tmRNA tagging, meaning that the signal may not come from ribosome stalling on damaged mRNAs. The authors try to use rifampicin to block transcription, but it doesn't fully work. To get around this limitation, they measure tmRNA activity in S30 extracts with a reporter mRNA. They show that chemically damaged mRNA leads to higher levels of tmRNA tagging of reporter protein than untreated mRNA. The data in Figure 7 convincingly tie tagging to mRNA damage (and not rRNA or tRNA damage) and rule out concerns about premature transcriptional termination making non-stop mRNAs. This is a significant improvement in the paper.

4) A nice feature of the Zaher lab's earlier paper in 2018 (Nat. Comm.) about nucleotide damage and ribosome rescue in yeast was the link to RNA decay. To more conclusively show that tmRNA is acting on messages with alkylated damage, the authors could perform a pulse chase experiment, first labeling mRNA (the pulse) and then during the chase period, adding a short alkylation treatment, then observing the rates of turnover of labeled mRNA.

The authors addressed this concern by adding Figure 6—figure supplement 1 using antibodies to detect modified nucleotides in RNAs. The presentation of these data is confusing, but I take it that there are three samples, untreated, MMS treated, and a 10 min chase after MMS treatment. It requires an additional base treatment to see the isomerization required to detect m1A using an m6A antibody (a clever trick of nucleoside chemistry). The authors argue that after 10 min of chase, the signal drops in the WT more than in the ssrA knockout. But the signal in the 0 and 10 lanes of the WT look pretty similar to me, as do the 0 and 10 lanes of the ssrA knockout (except that there is more signal in the ssrA lanes). I am not fully convinced that these data tell a compelling story about mRNA decay. But they could be used to show that there are higher levels of m1A in mRNAs in ssrA knockout cells. In the future, the authors might consider separating mRNA from rRNA and tRNA and performing LC/MS to measure m1A levels. There are very efficient methods of removing rRNA from samples for sequencing (e.g. Illumina Ribo-Zero plus) that can likely be helpful.

We feel that the authors should soften their conclusion and not claim a change in rate of turnover based on the data in Figure 6—figure supplement 1; however, they can claim increased levels of modified mRNAs.

---

## [Author Response]

[Editors’ note: the authors resubmitted a revised version of the paper for consideration. What follows is the authors’ response to the first round of review.]

Reviewer #1:This paper from the Zaher lab describes studies examining the impact on translation caused by alkylation of mRNA caused by MMS or MNNG treatments. in vitro, m1A modification of the middle base of an A site codon, substantially impaired the rate and extent of peptide formation. Based on this defect in peptide bond formation, the researchers examined whether the stalled translation product was a substrate for rescue by the tmRNA system. First, they found that the m1A modification did not substantially impair trans-translation in vitro. Next, using a tagged tmRNA, they found that alkylative damage enhanced tagging of many cellular proteins in vivo, suggesting that alkylative damage generates tmRNA substrates in the cell. Finally, the authors found that disabling the trans-translation rescue pathway sensitized *E. coli* cells to alkylation of mRNAs.Overall, this is an interesting story and complements previous work showing that alkylative mRNA damage in eukaryotes triggers ribosome quality control (RQC) pathways including degradation of the stalled nascent peptide. The work in this paper indicates that a parallel quality control pathway is functional in bacteria to eliminate the stalled translation products caused by mRNA alkylation.Specific comments:1) In Figure 1A, why do the levels m6A decrease in the presence of MMS and MNNG?

This is a good observation, and we really do not have a good explanation for it. Our new immuno-blotting data in Figure 6—figure supplement 1 are in agreement with these observations. We note that these changes appear to be statistically insignificant, but that does not mean it is biologically insignificant too. A model that can explain these observations is that alkylative stress activates decay processes that target m^6^A-modified mRNAs. As this is all conjecture, and would distract from our discussion of the m1A effect, we decided not to elaborate on it.

2) In the last sentence of Results paragraph two, it is stated that the base level of m1G is at least 200-fold higher than that of m1A. However, this is not obvious based on Figures 1A and 1B and the different Y-axes used; in fact, it looks like m1A levels are 4-5-fold higher than m1G.

We apologize about the confusion, which we believe stems from referencing the wrong figure panels in the text (see next point). The m1A data is presented in Figure 1D and not Figure 1A. In the absence of MMS or MNNG, m1A levels are less than 20 pmole/mmole, whereas m1G levels are > 1000 pmole/mmole.

3) Results, third paragraph: the citations to different panels in Figure 1 are incorrect. Second line should refer to Figure 1D and 1E; third line to Figure 1F and panel C (m7G) is never cited in the text.

In the revised manuscript, this was fixed.

4) Figure 3: there is no positive control to show that the antibiotic was functional in these experiments

This is a good point. The control for these experiments is to add antibiotics to the unmodified complex, but since this a matched one, paromomycin has no effect. We opted not to include data showing the antibiotics work because these complexes would be irrelevant to the one shown in Figure 3. We also note that these experiments were all conducted at the same time as assays that we conducted on mismatched complexes published last year in Nucleic Acids Research.

5) Figure 4 is described as testing the 20 possible ternary complexes. However, the number of possible ternary complexes is not dictated by the amino acid, but by the tRNAs (around 46 different tRNAs). Since the experiment was examining the ability of different tRNAs to decode the modified codon, the number of possible tRNAs (with distinct anticodons) is more relevant than the different amino acids.

Good point. The revised manuscript now states that we reacted the initiation complexes with “all 20 aa-tRNA isoacceptors” instead.

6) Are the modified (alkylated) mRNAs turned over? Do the levels of modified nucleotides increase in the ∆-ssrA cells treated with MMS?

This was a great suggestion, and ideally we would have liked to use LC-MS to conduct these experiments. Notably, our published experiments in yeast relied on the polyA tail to enrich for mRNAs before they were analyzed for modifications. Obviously, this approach cannot be used in bacteria. Electroporating modified mRNAs into wild-type and DssrA cells to follow their decay would have been a good alternative. However, these experiments were also difficult to conduct in bacteria. Given these limitations, we ended up using immuno-blotting of gel-separated RNAs to look at the levels of modified nucleotides in wild-type and DssrA cells following an MMS challenge. These experiments allowed us to infer the level of modifications in mRNAs based on their migrations on the gel. These data are presented in Figure 6—figure supplement 1 and show that m1A-modified mRNAs appear to decay slower in the absence of tmRNA. We concede that these experiments are not ideal, but more direct experiments are nearly impossible to conduct in bacteria. Furthermore, in the context of all of the experiments presented in the manuscript, our data provide a compelling evidence for a role of tmRNA in damaged-mRNA quality control.

Reviewer #2:This manuscript by Zaher and co-workers describes the effect of alkylation agents on protein synthesis in E. coli. First they characterize which bases in total RNA are modified by the alkylating agents MMS and MNNG. An earlier report using in vitro extracts (You, Dai and Wang, 2017) showed that m1A and other modifications in mRNA severely reduce protein synthesis in E. coli. Here the authors use purified components and pre-steady state kinetic methods to demonstrate that the rate and overall yield of peptide bond formation is dramatically reduced when m1A is found at the second position of the A-site codon. There is a severe decoding defect as evidenced by the fact that addition of antibiotics known to induce miscoding with near-cognate (but not non-cognate) tRNAs do not improve decoding of a codon with an alkylated base. These biochemical data are compelling although they provide only a small step forward in terms of understanding how ribosomes deal with alkylated messages.The authors then ask how *E. coli* cells deal with ribosomes stalled on alkylated bases. They show that the activity of tmRNA, the main ribosome rescue factor, is increased when cells are treated with alkylating agents. This is an exciting finding because traditionally tmRNA has been known to work on truncated transcripts (lacking a stop codon); a role for ribosome rescue on chemically damaged mRNAs is new. My enthusiasm is tempered by three concerns: 1) they have not completely ruled out that alkylating agents damaging DNA might create truncated transcripts (the usual substrate for tmRNA) that could explain the increase in tmRNA activity without tying it to alkylation of mRNA.

We have taken this point to heart and tried to address it in our revised manuscript (see below).

2) There is a major gap in evidence and understanding between the in vitro biochemistry and the proteome-wide tmRNA activity detected in vivo.

Please see our response to this point below.

3) The physiological importance of tmRNA in responding to alkylating conditions seems to be quite minor: yes, there is a one hour lag in recovery when tmRNA is lost, but wild-type and tmRNA minus cells survive this treatment equally well.

We respectfully disagree with the reviewer. An hour-long delay for the DssrA would effectively dilute them about eightfold every time they are subjected to alkylative stress in a competitive environment where wild-type cells exist. As a result, it does not take many encounters with alkylation agents to dilute these cells away.

1) A major concern is transcription from damaged DNA could make truncated messages that increase tmRNA tagging, meaning that the signal may not come from ribosome stalling on damaged mRNAs. The authors try to use rifampicin to block transcription, but it doesn't fully work. The ciprofloxacin His6 signal is not eliminated by Rif pre-treatment and Ada is still being made: this means that transcription still is happening.An alternative strategy would be to work directly at the RNA level: electroporate mRNA encoding a reporter with a N-terminal FLAG tag. Pull down all the reporter protein of any length with anti-FLAG antibodies and see how much of it is tagged with His6 by the altered tmRNA. The mRNA could be treated with various amounts of alkylation prior to electroporation. This experiment would rule out any effects of DNA damage. It could be done in S30 extracts (where tmRNA is active) if electroporation proves to be impossible.

This was a great suggestion. As the reviewer alluded to, mRNA electroporation in bacteria is not easy. Nevertheless, we attempted these experiments multiple times using different conditions and different types of mRNAs and failed to observe any protein production. We even used phage mRNAs, hoping their increased stability will allow them to be translated efficiently after electroporation. The reviewer also suggested that if these experiments were difficult to conduct, S30-type experiments can be used as an alternative. This is exactly what we ended up doing, and the results are now shown in Figure 7. Briefly, the new data convincingly show that MMS-treated mRNA leads to tmRNA tagging in the extracts.

2) One of the nice features of the Zaher lab's earlier paper in 2018 (Nat. Comm.) about nucleotide damage and ribosome rescue in yeast was the link to RNA decay. In bacteria, tmRNA-mediated ribosome rescue leads to mRNA degradation by RNase R. In contrast, in the absence of tmRNA, rescue by ArfA does not promote mRNA degradation. To more conclusively show that tmRNA is acting on messages with alkylated damage, the authors could perform a pulse chase experiment, first labeling mRNA (the pulse) and then during the chase period, adding a short alkylation treatment, then observing the rates of turnover of labeled mRNA. They could compare the rates in wild-type cells vs those lacking tmRNA or RNase R. This experiment would rule out transcription problems (since the RNA is synthesized before alkylating agents are added). It might also show that alkylation affects mRNA decay and tie tmRNA activity to rescue on these transcripts by its effects on their decay.

Again, this is a nice suggestion by the reviewer. Please see our response to reviewer 1 above.

Reviewer #3:This study from the Zaher lab addresses the possible effects of alkylating agents on translation, in particular as a result of m1A adducts generated in mRNA. The in vitro and in vivo assays show significant effects on translation, and the authors also investigated how tmRNA might be part of the cellular response to alkylation. The study is fairly interesting, if not entirely unsurprising in its findings. The role of tmRNA is based on prediction rather than a discovery-based approach, resulting in the study providing an ultimately limited insight into the cellular response to alkylation at the level of translation.

We agree with the reviewer that the finding that tmRNA is activated in response to alkylation stress is expected given that alkylation to mRNA stalls the ribosome. However, we respectfully disagree with their assessment that the prediction-based approach makes the paper less impactful. Most science is prediction-based. More importantly, to the best of our knowledge, this is the first report that shows a translation-based quality control in bacteria is activated in response to alkylation stress.

Numbered summary of any substantive concerns1) Why were "sub-optimal substrates" used for the in vitro trans translation assays?

This is a good point. Making these modified mRNAs is not easy or cheap, so we opted not to make one specifically to study tmRNA. We should also note that as reviewer 2 alluded to, there is no reason to believe that methylation would affect tmRNA function.

2) The possible roles of ArfA and ArfB were not experimentally investigated, it is possible they also protect against m1A adducts (as is mentioned in the Discussion) or is this a tmRNA-specific effect? Some simple growth assays in the appropriate genetic backgrounds might be informative.

We suspect that ArfA and ArfB to play a role during alkylation (we briefly discussed this in the paper), but in this manuscript we wanted to focus on trans translation given its conservation in bacteria and that it is the primary process for ribosome rescue.

3) As the authors concede for the in vivo experiments, possible effects of damage to the ribosomes cannot be excluded. The argument in the discussion is not convincing as to alkylation could not be assessed, if only globally initially, for 30S and 50S subunits.

We believe the new data with the S30 extract addresses some of these concerns (described above). In these experiments, only mRNA is damaged before its addition to the S30 extract, showing robust activation of tmRNA.

[Editors’ note: what follows is the authors’ response to the second round of review.]

Revisions:1) It is striking that the pattern of tma-tagged products is not altered upon MMA treatment. This is noted in vivo in Figure 6 where the level abundance of tagged products are increased in MMS-treated versus untreated cells, but the pattern of products on the gel are not changed. The new data in Figure 7 shows the same result in vitro. The untreated and treated mRNAs yield the same pattern of tma-tagged products. The authors propose that this is due to the specificity of the antibody and they show that different antibodies reveal different patterns. While the authors' hypothesis seems reasonable at this time, and no additional experiments are requested, we feel that this unexpected result is commented on in this review.

Point well taken, and we have included a discussion of these observations in the Discussion session of the revised manuscript. In particular we added the following text “Future experiments aimed at characterizing the identity of the tagged peptides, using mass-spectrometry approaches for example, are likely to shed important insights into the specificity of tmRNA tagging as well as that of the antibody recognition of tagged products. Since MMS is known to modify A and C, at least in a way that affects their Watson-Crick-base pairing properties, we expect tagging to occur preferentially on codons enriched for these two nucleotides.”

2) Is it appropriate to refer to Figure 2 as testing "all 20 possible aa-tRNA isoacceptors"? While perhaps formally correct, they did test one isoacceptor for each amino acid, the relevant tRNA attribute in the assay is the anticodon. I would be more comfortable if they stated that they tested 20 different tRNAs with unique anticodons. Later in the paragraph, they state that "no significant dipeptide accumulation occurred in the presence of any aa-tRNA". I think this should read "no significant dipeptide accumulation occurred in the presence of any of the tested aa-tRNAs"

In this experiment we aminoacylate bulk tRNA with one amino acid and the corresponding tRNA synthetase at a time. For amino acids that have different isoacceptors, all of them will be charged regardless of their anticodon identity. So for each PT reaction, all isoacceptors for that specific amino acid can in principle react with our initiation complex. We hope this clarifies some of the confusion.

3) One concern is that transcription from damaged DNA could make truncated messages that increase tmRNA tagging, meaning that the signal may not come from ribosome stalling on damaged mRNAs. The authors try to use rifampicin to block transcription, but it doesn't fully work. To get around this limitation, they measure tmRNA activity in S30 extracts with a reporter mRNA. They show that chemically damaged mRNA leads to higher levels of tmRNA tagging of reporter protein than untreated mRNA. The data in Figure 7 convincingly tie tagging to mRNA damage (and not rRNA or tRNA damage) and rule out concerns about premature transcriptional termination making non-stop mRNAs. This is a significant improvement in the paper.

We thank the reviewers for this comment.

4) A nice feature of the Zaher lab's earlier paper in 2018 (Nat. Comm.) about nucleotide damage and ribosome rescue in yeast was the link to RNA decay. To more conclusively show that tmRNA is acting on messages with alkylated damage, the authors could perform a pulse chase experiment, first labeling mRNA (the pulse) and then during the chase period, adding a short alkylation treatment, then observing the rates of turnover of labeled mRNA.The authors addressed this concern by adding Figure 6—figure supplement 1 using antibodies to detect modified nucleotides in RNAs. The presentation of these data is confusing, but I take it that there are three samples, untreated, MMS treated, and a 10 min chase after MMS treatment. It requires an additional base treatment to see the isomerization required to detect m1A using an m6A antibody (a clever trick of nucleoside chemistry). The authors argue that after 10 min of chase, the signal drops in the WT more than in the ssrA knockout. But the signal in the 0 and 10 lanes of the WT look pretty similar to me, as do the 0 and 10 lanes of the ssrA knockout (except that there is more signal in the ssrA lanes). I am not fully convinced that these data tell a compelling story about mRNA decay. But they could be used to show that there are higher levels of m1A in mRNAs in ssrA knockout cells. In the future, the authors might consider separating mRNA from rRNA and tRNA and performing LC/MS to measure m1A levels. There are very efficient methods of removing rRNA from samples for sequencing (e.g. Illumina Ribo-Zero plus) that can likely be helpful.We feel that the authors should soften their conclusion and not claim a change in rate of turnover based on the data in Figure 6—figure supplement 1; however, they can claim increased levels of modified mRNAs.

This is a very good point, and we softened our language in the revised manuscript in regard to this point. The relevant section now reads “More important was the observation that the MMS-induced signal was much higher in the DssrA cells (Figure 6—figure supplement 1). Our observations suggest that not only does tmRNA rescue ribosomes stalled on modified mRNAs, but that in its absence m1A levels in mRNA increase.”